

# Quantum echo dynamics in the Sherrington-Kirkpatrick model

Silvia Pappalardi [1,2,3,⋆], Anatoli Polkovnikov [3] and Alessandro Silva [2]

**1** SISSA — International School for Advanced Studies,
via Bonomea 265, I-34136 Trieste, Italy
**2** Abdus Salam ICTP — International Center for Theoretical Physics,
Strada Costiera 11, I-34151 Trieste, Italy
**3** Department of Physics, Boston University,
590 Commonwealth Avenue, Boston, Massachusetts 02215, USA

⋆ spappala@sissa.it

## Abstract

Understanding the footprints of chaos in quantum-many-body systems has been under debate for a long time. In this work, we study the echo dynamics of the Sherrington-Kirkpatrick (SK) model with transverse field under effective time reversal. We investigate numerically its quantum and semiclassical dynamics. We explore how chaotic many-body quantum physics can lead to exponential divergence of the echo of observables and we show that it is a result of three requirements: i) the collective nature of the observable, ii) a properly chosen initial state and iii) the existence of a well-defined chaotic semi-classical (large-$N$) limit. Under these conditions, the echo grows exponentially up to the Ehrenfest time, which scales logarithmically with the number of spins $N$. In this regime, the echo is well described by the semiclassical (truncated Wigner) approximation. We also discuss a short-range version of the SK model, where the Ehrenfest time does not depend on $N$ and the quantum echo shows only polynomial growth. Our findings provide new insights on scrambling and echo dynamics and how to observe it experimentally.

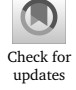

# 1   Introduction

Understanding how irreversibility arises in classical and quantum systems has been of pivotal importance since the foundations of statistical mechanics [1–6]. One of the most widely used ways to characterise chaotic dynamics in the quantum domain is the study of imperfect time-reversal evolution of the wave function, in particular, the *Loschmidt echo* [4]. Under classical chaotic dynamics, as a result of the exponential sensitivity of trajectories to small perturbations, any imperfection in a time-reversed protocol hinders a full recovery of the initial information, making time-reversal impossible in practice. Analogous approaches have been explored successfully in few-body quantum systems [7–9] but, as far as many-body systems are concerned, the onset of chaos is still the focus of an intense debate [10–12].

The topic was recently revived with a new name: *scrambling* [13–18]. This revival was mostly motivated by Kitaev's proposal to quantify chaos in many-body systems in terms of the growth in time of the square the non-equal time commutator of two initially commuting observables [15], or of the closely connected out of time order correlators (OTOC). These objects are defined as multi-point and multi-time correlation functions [1] which cannot be represented on a single Keldysh contour [19]. OTOC are characterized by an unusual time-ordering which prevents them from appearing in standard causal response functions. In the semiclassical limit, OTOC are generally related to the Lyapunov instabilities of classical trajectories, as such, they are good indicators of irreversibility [16–18].

Much before scrambling, these questions were addressed in the context of *nuclear magnetic resonance* (NMR) "magic echo" experiments [20–23], where irreversibility is characterized via the macroscopic response of some physical observable under imperfect time-reversal. A first attempt to analyze chaos of many-body quantum systems through the echo was made by B. Fine and collaborators in Refs. [24] and [25] for particular spins systems. There, it was found that the exponential sensitivity of the echo only applies to quantum systems close to their classical limit. More recently, it has been understood that the echo of observables is intimately linked to the square commutator and to OTOC, see also Refs. [26–29] for related

---

[1]More or equal than three body.

developments.

In the classical limit, the square of a non-equal time commutator of two observables maps to the square of the non-equal time Poisson bracket of the corresponding classical functions [16–18]. Thus, the expectation value of the square commutator over the initial quantum state corresponds to the averaging of the corresponding square Poisson bracket over the initial probability distribution, e.g. given by the Wigner function. Therefore, one anticipates that at least near the classical limit, the square commutator should grow exponentially fast in time with a rate given by the maximal Lyapunov exponent (similar considerations apply to the echo). Indeed, examples of quantum exponential sensitivity have been found only in models of few-particle systems near well-defined semi-classical limits [30–39] and in the large-$N$ limit of a Sachdev-Ye-Kitaev (SYK) model, a solvable model of all-to-all interacting fermions [15, 29, 40]. On the other hand, following the initial observation of Ref. [24] it was proven that for spin or fermionic systems with local interactions the OTOC of local observables or sums of local observables grows at most polynomially in time [41]. However, to date, the mechanisms that underpin the above footprints of chaos in many-body quantum systems are not fully understood.

In this work, we investigate how chaotic many-body quantum dynamics leads to the exponential divergence of the echo of observables in the transverse Sherrington-Kirkpatrick (SK) spin model with long-range interactions. This model can be experimentally realised over different atomic platforms ranging from cavity QED to Rydberg atoms, where it has been proposed as a way to access scrambling via interferometry [42]. Theoretically it has many analogies with the SYK model, as it shares the feature of having non-local all-to-all random interactions. At the same time, there are important differences between them: while the SK displays a quantum phase transition towards a quantum glass phase below a critical transverse field [43–45], the SYK model remains critical and scale invariant at all temperatures. Because of nonuniform couplings, the SK model can not be mapped to that of a large spin and for this reason there is no simple classical limit, in a way similar to the situation in the SYK model. Yet we show both analytically (c.f. Appendix B) and numerically (c.f. Sec. 5.1) that a semiclassical expansion such as the truncated Wigner approximation (TWA) [46–49] can accurately reproduce both the forward evolution of observables like magnetization (essentially up to infinitely long times) and the echo and hence the OTOC up to the Ehrenfest time. In this sense, even in the absence of a clear mapping to a classical Hamiltonian, the large $N$ limit of the SK model is semiclassical, similarly to the SYK model [29].

The availability of a chaotic semiclassical limit is also in this case the most important ingredient to see an exponential growth of the echo of observables (and of the OTOC). As in recent studies of the SYK model, we find that in order to have exponential behaviour of the OTOC it is necessary to have long-range interactions in the system, correctly captured by semiclassical TWA dynamics. In this way, even in the absence of an obvious classical limit in the system, $1/N$ serves as an effective Planck's constant $\hbar$. In order to further confirm the crucial role of a well-defined semi-classical limit, we also considered a short-range version of the SK model with random couplings between sites decaying gaussianly as a function of their distance. In this case, the semi-classical description fails to correctly reproduce the echo dynamics, which do not show exponential sensitivity to the protocol time. Our work therefore confirms that the existence of quantum Lyapunov exponents is closely related to the proximity of the model to the semiclassical limit, coinciding with the corresponding classical exponents [30–34] (c.f. Ref. [29] for the SYK model).

We also find that the nature of the initial state and of the observable are crucial to observe exponential echo response in this and other large $N$ models. In order to see exponential growth of the OTOC, the operator on the initial state has, in loose terms, to give enough "space" to the OTOC to develop an exponential growth. This means quantitatively that the intermediate

time window separating the early perturbative power law growth of the latter and its eventual saturation at long times has to be long and eventually divergent in the thermodynamic limit. This is clearly impossible in quantum systems with a bounded local Hilbert space size like in spin 1/2 chains or Hubbard like models of interacting fermions if we choose observables which are local in space. Such operators are bounded by the corresponding finite operator norms at long times and generically do not give room for exponential growth. In Ref. [41] it was thus argued that collective observables such as the sums of local observables, which can become arbitrarily large with the number of degrees of freedom, are better candidates for observing universal, non-perturbative behavior of OTOC. Thus, given a collective observable, one has to require that the long-time saturation value of the OTOC has a parametrically larger value in the system size $N$ than the coefficient governing the initial perturbative short-time behaviour. Interestingly, such requirements simultaneously constrain the nature of the initial state and of the observable. In particular, we find that for a collective observable the "good" initial state must be such that there is an extensive difference between the initial and the equilibrium (long time) values of the observable. For example, if we choose the total (non-conserved) magnetization as an observable, that decays to zero under forward evolution, one could start from an initially magnetized state. In the case of the current, a good initial state will be the one with a macroscopic current, and so on.

Such initial states naturally generalize those proposed by Rozenbaum et al. in Ref. [50], where the authors associated the existence of an exponential regime with the choice of the "classical" initial conditions localized in phase space, where the position and the momentum of the particle acquire non-zero expectation values. This choice of initial conditions is very similar to that proposed in Ref. [24] for studying the echo based on more intuitive considerations. Notice that in Ref. [40] it was argued that classical Lyapunov exponents can exceed the quantum one in SYK model. However, the authors of that paper considered initial conditions sampled according to the classical thermal Gibbs distribution rather than the corresponding Wigner function. The two choices can lead to inconsistencies between the conclusions, since the exponential growth crucially depends on the initial state.

The rest of the paper is organized as follows. In Section 2 we introduce the echo operator and discuss the connection between echo dynamics and scrambling. In Section 3 we discuss the requirements on the initial state and the observables. Then, in Section 4 we describe the Sherrington-Kirkpatrick (SK) model and its short-range version. In the following Section 5, we summarize the main aspects of the TWA and show its validity for the SK model in the thermodynamic limit and its failure for the short-range case. Finally in Section 6, we show numerical results and determine the Lyapunov exponent for the long-range model.

## 2 Echo dynamics and scrambling

Let us start with a description of the protocol we are studying: an imperfect time reversal through the echo response of an observable, in a spirit similar to Loschmidt echo experiments and to NMR magic echoes (see Refs. [24] and [29]). In this setting, the system is prepared in the eigenstate of some observable $\hat{A}$, such as a polarized state of the magnetization, and it is then allowed to evolve under the action of the Hamiltonian $-\hat{H}$ for a certain time $t$. At time $t$, it is then subject to a rapid rotation generated by the unitary operator $e^{i\epsilon\hat{B}}$ and it then evolves back under the reversed Hamiltonian $+\hat{H}$ for an identical time interval $t$. Afterwards,

the observable $\hat{A}$ is measured. The corresponding time-evolved operator $\hat{A}$ reads: [29]

$$
\begin{aligned}
\hat{A}_\epsilon(t) &= e^{i\hat{H}t/\hbar}e^{-i\epsilon\hat{B}}e^{-i\hat{H}t/\hbar}\hat{A}e^{i\hat{H}t/\hbar}e^{i\epsilon\hat{B}}e^{-i\hat{H}t/\hbar} = e^{-i\epsilon\hat{B}(t)}\hat{A}e^{i\epsilon\hat{B}(t)} \\
&= \hat{A} - i\epsilon\,[\hat{B}(t),\hat{A}] - \frac{\epsilon^2}{2}\,[\hat{B}(t),[\hat{B}(t),\hat{A}]] + \mathcal{O}(\epsilon^3)\,,
\end{aligned}
\tag{1}
$$

where $\hat{B}(t) = e^{i\hat{H}t/\hbar}\hat{B}\,e^{-i\hat{H}t/\hbar}$ is the perturbing operator in the Heisenberg representation with respect to the Hamiltonian $\hat{H}$.

The expectation value of the difference $\hat{A}_\epsilon(t) - \hat{A}$ on a generic quantum state $|\psi_0\rangle$ corresponds to the *echo response of the observable $\hat{A}$*. The term proportional to $\epsilon$ appears in a standard Kubo-type linear response susceptibility and does not contain information about unusual time-ordering. In general it should be subtracted from the echo. If the initial state $|\psi_0\rangle$ is an eigenstate of $\hat{A}$ this term vanishes and the leading order term of the difference is the second one [29], proportional to an OTOC, i.e. a correlator without a causal structure that therefore cannot appear in response functions (see e.g. Ref. [19, 49, 51]). From now on we will deal only with such states. It is thus useful to define $\mu(t)$, characterizing the echo, as

$$
\mu(t) = \lim_{\epsilon\to0}\frac{1}{\epsilon^2}\langle\hat{A}_\epsilon(t) - \hat{A}\rangle_0 = -\frac{1}{2}\langle[\hat{B}(t),[\hat{B}(t),\hat{A}]]\rangle_0\,,
\tag{2}
$$

where $\langle\ldots\rangle_0$ stands for the average with respect to the initial state. The function $\mu(t)$ contains an OTOC as, in particular, it contains $\langle\hat{B}(t)\hat{A}\hat{B}(t)\rangle$.

Let us note that *the square commutator* $c(t) = -\langle[\hat{B}(t),\hat{A}]^2\rangle$ [15–18, 30–33, 40], in this language corresponds to the second moment of $\hat{A}_\epsilon(t) - \hat{A}$ computed with Eq.(1) at second order in $\epsilon$ (see Ref. [52, 53] for a related discussion). It is well known that the classical limit of $c(t)$ encodes the square of the derivatives of the classical trajectory to respect to the initial conditions [16–18]. Thus, whenever the classical limit is chaotic, $c(t)$ is expected to grow exponentially in time. While the square commutator is generally different from $\mu(t)$, as it contains a different OTOC, in the semi-classical limit both expressions have a similar structure containing the square of the derivatives of trajectories with respect to the initial conditions, which grow exponentially in time with the corresponding Lyapunov exponent in the presence of the semi-classical chaos (c.f. Refs. [7,24,29]). In Appendix A, we derive this result formally, by computing the semi-classical limits of the echo observable and of the square commutator using the Bopp representation of the operators [49]. A very interesting and open question concerns the distribution of the echo operator $\hat{A}_\epsilon(t) - \hat{A}$. We will leave this study for future work and focus here only on studying its expectation value $\mu(t)$.

## 3 The choice of the initial state and the observable

The typical time dependence of the echo of observables (as well as that of other OTOCs) is divided into three regimes: an initial perturbative one, reaching times of the order of the inverse coupling constant, where the echo grows as a power law, and an eventual saturation at long-times (beyond the Ehrenfest time) separated by an intermediate regime, where the presence of quantum chaos is manifest as an exponential growth. It is clear that, in order for such exponential behaviour to be seen, the long-time saturation value of the echo has to be parametrically larger in the system size $N$ than the coefficient governing its initial perturbative short-time expansion. This requirement puts some well defined constraints on the type of observables and of initial states to be considered.

Let us now explain quantitatively this point by first analyzing the short-time regime with perturbation theory and then the long-time saturation value, evaluated with the Eigenstate

Thermalization Hypothesis (ETH) [54, 55]. We will show in the generic case of a sufficiently chaotic spin Hamiltonian satisfying ETH, that the conditions above are met if one chooses i) either the perturbation $\hat{B}$ or the observable $\hat{A}$ to be collective (sum of local operators), ii) the initial expectation value of $\hat{A}$ in the state $|\psi_0\rangle$ far from the long-time (thermal) saturation value. As we already mentioned above, while it is not required the analysis significantly simplifies if the initial state is the eigenstate of $\hat{A}$

$$|\psi_0\rangle = |\alpha_0\rangle \quad : \quad \hat{A}|\alpha_0\rangle = \alpha_0 |\alpha_0\rangle . \tag{3}$$

## 3.1 Early-time growth

Let us start with the initial growth. Using Eq.(3), the average of the echo operator in Eq.(2) becomes

$$\mu(t) = \langle \hat{B}(t)\hat{A}\hat{B}(t)\rangle - \alpha_0 \langle \hat{B}^2(t)\rangle . \tag{4}$$

In order to derive the early-time behaviour, it is more convenient to work in the eigenbasis of the operator $\hat{A}$, i.e. $\hat{A}|\alpha_\lambda\rangle = \alpha_\lambda |\alpha_\lambda\rangle$ with $\lambda = 0, \dots, D-1$, where $D$ is the Hilbert space dimension ($D = 2^N$ for a system of $N$ spins $1/2$). The early-time expansion of the operator $\hat{B}(t)$ can be obtained via the Baker–Campbell–Hausdorff formula. Up to second order in time it reads

$$\hat{B}(t) = \hat{B} - it[\hat{H}, \hat{B}] - t^2/2[\hat{H}, [\hat{H}, \hat{B}]] + \mathcal{O}(t^3) . \tag{5}$$

We will further assume that the operators $\hat{A}$ and $\hat{B}$ commute at $t = 0$. This guarantees that $\mu(0) = 0$, i.e. that the echo signal in $\hat{A}$ only appears after some propagation time. For this reason the operators $\hat{A}$ and $\hat{B}$ can be simultaneously diagonalized such that $\hat{B}|\alpha_\lambda\rangle = \beta_\lambda|\alpha_\lambda\rangle$. At short times, the average of the echo operator (4) reads

$$\mu(t) = t^2 \sum_{\lambda \neq 0} |\hat{H}_{0\lambda}|^2 (\beta_\lambda - \beta_0)^2 (\alpha_\lambda - \alpha_0) + \mathcal{O}(t^4) , \tag{6}$$

where $|\hat{H}_{0\lambda}| = \langle \alpha_0|\hat{H}|\alpha_\lambda\rangle$ are the matrix elements of Hamiltonian matrix elements in the eigenbasis of $\hat{A}$.

## 3.2 Long time saturation

Let us now turn to the analysis of the long time saturation of the echo, or more precisely of the infinite time average of Eq.(2)

$$\bar{\mu} = \lim_{T \to \infty} \frac{1}{T} \int_0^T \mu(t)\, dt .$$

Now it is convenient to work in the eigenbasis of the Hamiltonian, i.e. $\hat{H}|E_n\rangle = E_n|E_n\rangle$. Then Eq. (4) can be re-written as

$$\mu(t) = \sum_{nmpq} c_n c_q^* B_{nm} A_{mp} B_{pq}\, e^{i(E_n - E_m + E_p - E_q)t} - \alpha_0 \sum_{nmp} c_n c_m^* B_{np} B_{pm}\, e^{i(E_n - E_m)t} , \tag{7}$$

where $c_n = \langle \psi_0|E_n\rangle$, $B_{nm} = \langle E_n|\hat{B}|E_m\rangle$ and $A_{nm} = \langle E_n|\hat{A}|E_m\rangle$. We will assume that the Hamiltonian $\hat{H}$ is chaotic satisfying ETH and in particular that it has no degeneracies. With this choice, the time average of Eq.(7) is non-zero only if the energies appearing in the exponentials are equal to each other pairwise, [54, 55] such that

$$\overline{e^{i(E_n - E_m + E_p - E_q)t}} = \delta_{nm}\delta_{pq} + \delta_{nq}\delta_{mp} - \delta_{nmpq} ,$$

where $\delta_{nmpq}$ implies that all four indices are equal to each other. Likewise

$$\overline{e^{i(E_n - E_m)t}} = \delta_{nm} \ .$$

Then

$$\overline{\mu} = \sum_{nm} c_n c_m^* B_{nn} A_{nm} B_{mm} + \sum_{nm} |c_n|^2 |B_{nm}|^2 A_{mm} - \sum_n |c_n|^2 B_{nn}^2 A_{nn} - \alpha_0 \sum_{nm} |c_n|^2 |B_{nm}|^2$$

$$= \sum_{nm} c_n c_m^* B_{nn} A_{nm} B_{mm} - \alpha_0 \sum_n |c_n|^2 |B_{nn}|^2 + \sum_{n \neq m} |c_n|^2 (A_{mm} - \alpha_0) |B_{nm}|^2 \ . \tag{8}$$

This expression further simplifies if we assume that the diagonal matrix elements $B_{nn}$ are smooth functions of $E_n$, an assumption always justified within ETH. If indeed the energy fluctuations of the initial state $\delta^2 E = \langle \psi_0 | \hat{H}^2 | \psi_0 \rangle - \langle \psi_0 | \hat{H} | \psi_0 \rangle^2$ are sub-extensive $\delta E^2 / E^2 \sim 1/N$ [55], owing to the fact that $\sum_{nm} c_n c_m^* A_{nm} = \alpha_0$ and $\sum |c_n|^2 = 1$, the first two terms in the expression above cancel each other and we get

$$\bar{\mu} \approx \sum_{n \neq m} |c_n|^2 (A_{mm} - \alpha_0) |B_{nm}|^2 \ . \tag{9}$$

We can now compute the long-time saturation value using the ETH ansatz for the matrix elements of observables in the eigenbasis of the Hamiltonian. The latter is formally stated as [54, 55]

$$A_{nm} = \mathcal{A}(\bar{E}) \delta_{nm} + e^{-S(\bar{E})/2} f_{\hat{A}}(\bar{E}, \omega_{nm}) R_{nm}, \tag{10}$$

where $\bar{E} = (E_n + E_m)/2$, $\omega_{nm} = E_m - E_n$, $S(\bar{E})$ is the micro-canonical entropy and $R_{nm}$ is a random variable with zero average and unit variance. Both $\mathcal{A}(\bar{E})$ and $f_{\hat{A}}(\bar{E}, \omega_{nm})$ are smooth functions of their arguments. We can now substitute it into E.(9) and obtain

$$\bar{\mu} = \sum_{n \neq m} |c_n|^2 [\mathcal{A}(E_n + \omega_{nm}) - \alpha_0] \left| f_{\hat{B}}(E_n + \omega_{nm}/2, \omega_{nm}) \right|^2 e^{-S(E_n + \omega_{nm}/2)} \ , \tag{11}$$

where we have replaced $R_{mm}$ ($|R_{nm}|^2$) with its statistical zero (unit) average and $\bar{E} = E_n + \omega_{nm}/2$ and $E_m = E_n + \omega_{nm}$. We now write each sum as an integral with the suitable density of states, $\sum_m \to \int_0^\infty dE_m\, e^{S(E_m)} = \int d\omega\, e^{S(E + \omega)}$. We therefore have

$$\bar{\mu} = \sum_n |c_n|^2 \int d\omega\, [\mathcal{A}(E_n + \omega) - \alpha_0] \left| f_{\hat{B}}(E_n + \omega/2, \omega) \right|^2 e^{-S(E_n + \omega/2) + S(E_n + \omega)} \ . \tag{12}$$

Since $f_{\hat{B}}(E, \omega)$ decays rapidly enough at large $\omega$ [56], we can expand in powers of $\omega$

$$\mathcal{A}(E_n + \omega) = \mathcal{A}(E) + \frac{\partial \mathcal{A}}{\partial E} \omega + \dots \tag{13a}$$

Notice that if $\hat{A}$ is a local operator, or a sum of local operators, the term containing the energy derivative become irrelevant in the thermodynamic limit [55]. Substituting back, we obtain

$$\bar{\mu} = \sum_n |c_n|^2 [\mathcal{A}(E_n) - \alpha_0] \int d\omega \left| f_{\hat{B}}(E_n + \omega/2, \omega) \right|^2 e^{-S(E_n + \omega/2) + S(E_n + \omega)}$$

$$= \sum_n |c_n|^2 [\mathcal{A}(E_n) - \alpha_0] \langle E_n | \Delta \hat{B}^2 | E_n \rangle \ , \tag{14}$$

where we have replaced the frequency integral by the variance over a single energy eigenstate $\langle E_n | \Delta \hat{B}^2 | E_n \rangle = \langle E_n | \hat{B}^2 | E_n \rangle - \langle E_n | \hat{B} | E_n \rangle^2$, see Ref. [55]. Performing now an expansion around the average energy $E = \langle \psi_0 | \hat{H} | \psi_0 \rangle$

$$\mathcal{A}(E_n) = \mathcal{A}(E) + (E_n - E)\,\mathcal{A}'(E) + \frac{1}{2}\,(E_n - E)^2\,\mathcal{A}''(E) + \dots, \qquad (15)$$

where $\mathcal{A}'(E) = \frac{\partial \mathcal{A}}{\partial E}|_E$ and $\mathcal{A}''(E) = \frac{\partial^2 \mathcal{A}}{\partial E^2}|_E$. One then obtains

$$\bar{\mu} = (\mathcal{A}(E) - \alpha_0)\Delta B^2(E) + \delta E^2 \left[ (\mathcal{A}(E) - \alpha_0)(B'(E))^2 + \frac{1}{2}\mathcal{A}''(E)\Delta B^2(E) \right], \qquad (16)$$

where we isolated the corrections proportional to $\delta E^2$. If $\hat{B}$ is a local operator, these corrections are suppressed by a factor of $N$ compared to the first leading term. On the other hand, when $\hat{B}$ is a sum of local operators, then the correction (proportional to $B'(E)$) scales with $N$ in the same way as the first leading term.

### 3.3 Existence of a parametric window for the echo growth

We are now in the position to compare the short and the long-time behavior and find the conditions under which there is a parametric window for the growth of the echo. A simple qualitative criterion, which is at the same time a necessary condition, for the existence of such a window is

$$|\mu(t^*)| \sim N^{-\ell}|\bar{\mu}|,$$

where $\ell$ is a positive power and $t^*$ is the time of breakdown of the short time expansion. We will focus only on a class of operators $\hat{A}$ and $\hat{B}$ which are either local in spins or can be represented as sums of local terms, i.e. we will focus on most common and measurable operators representing physical observables. In addition, we will also assume that the Hamiltonian contains sums of few spins (fermion) terms, i.e. it can contain an external field and two or three spin interactions, which may not necessarily be local in space. Under these assumptions, the Hamiltonian can flip at most few spins. Therefore, for the states connected by the nonzero matrix element $|H_{0\lambda}|^2$, the differences $\alpha_\lambda - \alpha_0$ and $\beta_\lambda - \beta_0$ are non-extensive irrespective on whether $\hat{A}$ or $\hat{B}$ are local or sums of local terms. Therefore $|\alpha_\lambda - \alpha_0|$ and $|\beta_\lambda - \beta_0|$ are bounded by some non-extensive constants $M_A = \text{Max}_\lambda |\alpha_\lambda - \alpha_0|$ and $M_B = \text{Max}_\lambda |\beta_\lambda - \beta_0|$.

Let us start by estimating the short time expansion using Eq.(6) distinguish three different possibilities, which we discuss one by one: (i) both $\hat{A}$ and $\hat{B}$ are collective operators, (ii) one of the operators is global one is local and (iii) both $\hat{A}$ and $\hat{B}$ are local.

*(i) $\hat{A}$ and $\hat{B}$ are global operators.* In this case at short times

$$\begin{aligned}
\mu(t) &\leq t^2 \sum_{\lambda \neq 0} |\hat{H}_{0\lambda}|^2 |\beta_\lambda - \beta_0|^2 |\alpha_\lambda - \alpha_0| \\
&\leq t^2 M_A M_B^2 \sum_{\lambda \neq 0} |\hat{H}_{0\lambda}|^2 \sim C t^2 N,
\end{aligned} \qquad (17)$$

where we used the standard normalization of the Hamiltonian such that it has an extensive energy variance

$$\langle \psi_0 | \hat{H}^2 | \psi_0 \rangle = \sum_\lambda |\hat{H}_{0\lambda}|^2 \propto N.$$

We note that this scaling with $N$ can be reduced further if $\alpha_\lambda - \alpha_0$ has an alternating sign between different eigenstates $\alpha_\lambda$. The time $t^*$ defining the validity of the short time expansion

can be estimated from the decay of the expectation value of $\hat{B}(t)$, which is readily obtained from Eq. (5)

$$\langle\psi_0|\hat{B}(t)|\psi_0\rangle = \beta_0 + t^2 \sum_{\lambda} |H_{0\lambda}|^2 (\beta_\lambda - \beta_0) + O(t^3) \,.$$

By equating the first and the second term in the expansion and by the same arguments of extensivity of the energy variance in the initial state we see that the time $t^*$ is $N$-independent.

*(ii) One of the operators $\hat{A}$ or $\hat{B}$ is local and the other is extensive.* In this case locality of one of the operators $\hat{A}$ or $\hat{B}$ (let us say $\hat{B}$ for concreteness) restricts the eigenstates $|\alpha_\lambda\rangle$ in Eq. (17) to those where one of the local degrees of freedom (e.g. a spin) is localized. This additional selection rule removes a factor of $N$ from the the sum in Eq. (17) leading to the following estimate

$$\mu(t) \sim C t^2 \,. \tag{18}$$

It is easy to see that the time scale $t^*$ is $N$-independent irrespective of whether the operator $\hat{B}$ is extensive or global.

*(iii) Both $\hat{A}$ and $\hat{B}$ are local.* We will focus on operators that are not spatially separated. The OTOC for spatially separated operators was analyzed in the literature, see e.g. Ref. [57]. In these situations, there is a possibility for exponential echo growth, related to the out of the light cone dynamics and not generally connected to the existence of chaos. Assuming that there is no spatial separation, we can easily check that the Eq. (18) still holds.

Let us now determine the long time asymptotes of $\mu(t)$ from Eq. (9) for the three cases. As already mentioned, the scaling of these asymptotes with $N$ sets the condition for the initial state and operator. The best initial state to have the maximal room for the non-perturbative growth of the echo is such that the difference between the initial value and its long time limit $|\mathcal{A}(E) - \alpha_0|$ appearing in Eq.(16) is maximal. In the case of a global operator $\hat{A}$, the maximum possible difference is extensive $|\mathcal{A}(E) - \alpha_0| \propto N$; for a local operator the maximal possible difference is of the order of one: $|\mathcal{A}(E) - \alpha_0| \propto N^0$. Then we immediately find for Eq. (9) that for the case (i) $\bar{\mu} \propto N^2$. Likewise for the case (ii), i.e. when either $\hat{A}$ or $\hat{B}$ is an extensive operator we have $\bar{\mu} \propto N^1$ and finally for the case (iii) $\bar{\mu} \propto N^0$. Comparing these asymptotes with the short time expansions of $\mu(t)$ discussed above we see that in order to have a non-perturbative growth of echo one should chose either the possibility (i) or (ii), i.e. at least one of the two operators $\hat{A}$ or $\hat{B}$ should be extensive. In particular, a very convenient choice we will use most extensively below is (i) where $\hat{A} = \hat{B}$ are the global magnetization along a particular direction:

$$\hat{A} = \hat{S}^\alpha = \sum_{i=1}^{N} \hat{\sigma}_i^\alpha \quad \text{with} \quad \alpha = x, y, z \,. \tag{19}$$

This choice is analogous to the one used in Refs. [24] and [41] and with that of standard echo-experiments [20–23]. We will also show results for the other cases ((ii) and (iii)). Of course the existence of a parametric large in $N$ time window is only a necessary condition for the exponential growth of the echo (OTOC) but a not sufficient one. If, however, the dynamics in the large $N$ limit are semiclassical and chaotic then we generally expect a regime of exponential growth of $\mu(t)$. Conversely if in the large $N$ limit dynamics remains quantum, there is no a-priory reason to expect any exponential behaviour of $\mu(t)$. As we show below this is indeed the case in the SK model with local couplings, where the non-perturbative growth regime of $\mu(t)$ is a power law with a small non-integer exponent.

# 4 The Sherrington-Kirkpatrick model in transverse field

We will now corroborate our general discussion with an analysis of the Sherrington-Kirkpatrick (SK) model, describing a set of spins with infinite-range interactions in their $z$-components. To make this model dynamical we add a uniform transverse field. Below we will also analyze a version of this model with local interactions which decay in space according to a Gaussian law.

The Hamiltonian of the SK model in the transverse field reads

$$\hat{H} = -\frac{1}{2} \sum_{i \neq j}^{N} J_{ij} \hat{\sigma}_i^z \hat{\sigma}_j^z - h \sum_{i=1}^{N} \hat{\sigma}_i^x \, , \tag{20}$$

where $\hat{\sigma}_i^z$, $\hat{\sigma}_i^x$ are the Pauli matrices and the couplings $J_{ij}$ are random symmetric numbers distributed according to the Gaussian probability with zero mean and the variance $J^2/N$ as

$$J_{ij} = \frac{J}{\sqrt{N}} g_{ij} \, , \tag{21}$$

where $g_{ij}$ are Gaussian random numbers with zero average and unit variance. At equilibrium, the phase-diagram of the SK model has been extensively studied [43–45]. In the limit of zero transverse field ($h = 0$), one recovers the classical SK model, [58,59] which has a glass transition at the critical temperature $T_c = J$. The SK model in transverse field has a zero-temperature quantum phase transition at a critical magnetic field $h_c(T = 0) \sim 1.52J$. [44] Away from equilibrium, this model was explored in Refs. [60–63]. Recently, the SK model has been also considered in the context of scrambling in Ref. [42], as a natural setup to access the square commutator via interferometry in cold-atoms experiments. See also Refs. [64,65] for related models.

In what follows, we will also analyze a *short-range version* of the SK model. It is described by the same spin Hamiltonian (20), but the random couplings $J_{ij}$ connecting the sites $i$, $j$ now decay with the distance $r_{ij}$ according

$$J_{ij} = \frac{J}{\sqrt{N(\sigma)}} e^{-\frac{r_{ij}^2}{2\sigma^2}} g_{ij} \, , \tag{22}$$

where $\sigma$ is a parameter defining the interaction range. In one dimension with periodic boundary conditions the distance between any two sites is taken to be

$$r_{ij} = \min\left(|i - j|, N - |i - j|\right).$$

We normalized the couplings by the effective number of spins within the correlation length $\sigma$: $N(\sigma) = \sum_{i \neq j} e^{-r_{ij}^2/2\sigma^2}/N \sim \sqrt{\pi}/2 \, \sigma \, \mathrm{Erf}(N/\sigma)$. This choice correctly interpolates between the short range ($\sigma \approx 1$) and the long range ($\sigma \to \infty$) limits of the SK model, for example, always keeping the energy variance extensive in any factorizable state. In the infinite range limit $\sigma \to \infty$, the standard SK model is recovered (21) and $N(\sigma) = N$. In the opposite case when $\sigma \ll N$, the normalization is simply a constant $N(\sigma) \sim 2/\sqrt{\pi} \, \sigma$.

# 5 Semiclassical dynamics in the large $N$-limit: the truncated Wigner Approximation (TWA)

In order to connect the exponential growth of the echo with the availability of a semiclassical limit, we will combine exact diagonalization with the semi-classical *truncated Wigner approximation* (TWA) [46–49,66,67]. TWA naturally arises as a saddle point approximation to the

path integral representation of the time evolution of a generic observable on a Keldysh contour [49]. As we discuss in more detail in the Appendix B, TWA can be rigorously derived for the SK model in the large N-limit with $1/N$ serving as a proper saddle point parameter. For completeness, we briefly describe the implementation of the TWA method below. In the next section we outline the application of the TWA to the SK model.

The easiest way to derive the TWA for a spin system is to use Schwinger boson representation, where each spin $\hat{\vec{s}}_i$ is represented by two boson operators $\hat{a}_i$ and $\hat{b}_i$ for $i = 1, \ldots N$

$$\hat{s}_i^z = \frac{1}{2}(\hat{a}_i^\dagger \hat{a}_i - \hat{b}_i^\dagger \hat{b}_i) \quad \hat{s}_i^+ = \hat{a}_i^\dagger \hat{b}_i, \quad \hat{s}_i^- = \hat{b}_i^\dagger \hat{a}_i, \tag{23}$$

with the additional constraint that $\hat{a}_i^\dagger \hat{a}_i + \hat{b}_i^\dagger \hat{b}_i = 1$ for each site $i$. The dynamics of spins is equivalent to the dynamics of Schwinger bosons. Because this constraint is conserved in time for any spin Hamiltonian, it is sufficient to enforce it only in the initial density matrix. In this language one can formulate the path integral evolution of the observables using bosonic coherent states (see Appendix B and Ref. [49] for details). Then the bosonic fields $\hat{a}^\dagger$ and $\hat{a}$ (and similarly $\hat{b}^\dagger$ and $\hat{b}$) map to complex phase space variables $\boldsymbol{\alpha}^*$ and $\boldsymbol{\alpha}$ ($\boldsymbol{\beta}$ and $\boldsymbol{\beta}^*$), which have the conventional Poisson bracket relations: $\{\boldsymbol{\alpha}^*, \boldsymbol{\alpha}\} = i$ ($\{\boldsymbol{\beta}^*, \boldsymbol{\beta}\} = i$). Under this mapping any operator, including the density matrix, maps to a function of these variables known as the Weyl symbol, with the Weyl symbol of the density matrix termed as the Wigner function

$$\hat{O}(\hat{a}, \hat{a}^\dagger) \to O^w(\boldsymbol{\alpha}, \boldsymbol{\alpha}^*), \quad \hat{\rho}(\hat{a}, \hat{a}^\dagger) \to W(\boldsymbol{\alpha}, \boldsymbol{\alpha}^*).$$

Here $\boldsymbol{\alpha} = \{\alpha_j\}$ and $\boldsymbol{\alpha}^* = \{\alpha_j^*\}$ with the index $j$ going over both different Schwinger boson components and different spins. The TWA emerges as a saddle point approximation, here justified in the large N-limit, of the evolution of some observable $\hat{O}$ in the Schwinger-Keldysh path integral (see Appendix B and Refs. [49,68]) and reads:

$$\langle \hat{O}(t) \rangle = \mathrm{Tr}[\hat{\rho}_0 \hat{O}(t)] \simeq \int d\boldsymbol{\alpha}_0 \, d\boldsymbol{\alpha}_0^* \, W(\boldsymbol{\alpha}_0, \boldsymbol{\alpha}_0^*) O^w(\boldsymbol{\alpha}(t), \boldsymbol{\alpha}^*(t)) \equiv \overline{\overline{O^w(\boldsymbol{\alpha}(t), \boldsymbol{\alpha}^*(t))}}, \tag{24}$$

where the double overline represents the average weighted with the initial Wigner function. The time evolution of $\boldsymbol{\alpha}(t)$ and $\boldsymbol{\alpha}^*(t)$ within the TWA is deterministic set by the classical Hamiltonian equations of motion:

$$i \frac{d\alpha_j}{dt} = \frac{\partial H^w(\boldsymbol{\alpha}, \boldsymbol{\alpha}^*)}{\partial \alpha_j^*}. \tag{25}$$

Going back from Schwinger bosons to classical angular momentum variables one recovers standard classical Hamiltonian equations for spin (angular momentum) variables:

$$\dot{s}_\alpha^j = \{s_\alpha^j, H^w(\vec{s})\} = \epsilon_{\alpha\beta\gamma} \frac{\partial H^w(\vec{s})}{\partial s_\beta^j} s_\gamma^j, \quad \leftrightarrow \quad \frac{d\vec{s}_j}{dt} = \frac{\partial H^w}{\partial \vec{s}_j} \times \vec{s}_j, \tag{26}$$

where now $j$ is the spin index, $\alpha, \beta, \gamma$ stand for $x, y, z$ spin components, and $\epsilon_{\alpha\beta\gamma}$ is the fully anti-symmetric Levi-Civita symbol. In a similar manner the TWA allows one to compute *multi-time correlation functions* via the use of Bopp operators, which involves evaluating non-equal time response functions on classical trajectories (see Ref. [49] and Appendix A).

We note that while formally the equations of motion for the Schwinger bosons coincide with the equations obtained using the Dirac's variational principle [69], TWA goes well beyond these approximations as it includes quantum fluctuations encoded in the Wigner function, which, in many cases, are essential for correctly describing the dynamics of the system. Only in the limit of an infinitesimally narrow Wigner function describing the initial state, TWA

reduces to the so called Dirac time-dependent variational principle [70]. Also, generally the variational principle completely fails in describing non-equal time correlation functions and can not be used, for example, to compute the echo of observables and the OTOC. On the other hand, unlike the conventional Keldysh diagrammatic technique, the derivation of TWA is not tied to the exponential Gibbs form of the initial density matrix nor it relies on assumptions of small nonlinearities [71].

In quantum systems with a well defined classical limit, like a particle in an external potential or a system of spins with large angular momentum, the TWA is known to asymptotically describe quantum echoes at short times [16, 31, 50, 72]. This approach breaks down eventually at the so-called *Ehrenfest time* $t_{\mathrm{Ehr}}$, when quantum interference effects between classical trajectories become significant [16, 30, 72, 73]. Interestingly, the TWA for the forward evolution of observables usually works for much longer times and some the error remains bounded for infinitely long times. This observation suggests that the quantum echo is a very sensitive probe defining the crossover time scale separating semiclassical and quantum time evolution regimes. This time-scale $t_{\mathrm{Ehr}}$ typically diverges as we approach the classical limit. In particular, for a particle in a chaotic potential, it is known to be

$$t_{\mathrm{Ehr}} = \frac{1}{2\lambda_{max}} \log \frac{1}{\hbar} \,, \tag{27}$$

where $\lambda_{max} > 0$ is the maximal Lyapunov exponent of the classical dynamics [74]. For nonlinear spins the role of $1/\hbar$ is played by the spin size $S$ and the Ehrenfest time diverges as $\log(S)$. This gives us direct information on the SK model for uniform couplings, i.e. $J_{ij} = 1/N$, where Hamiltonian reduces to the one of a single large-spin $S = N/2$ [49]. We find here that even when couplings are randomly distributed as in Eq.(21) the situation does not change qualitatively. As we show below numerically, providing additional analytical arguments in the Appendix B, also in this case the large $N$-limit ensures the validity of the saddle point approximation, hence of the TWA, with $1/N$ playing the role of the effective Planck's constant. Similar recent findings for the SYK model were reported in Ref [29]. In this sense the situation is similar to equilibrium, where the large $N$-limit ensures the validity of the saddle point mean-field approximation.

## 5.1 TWA for the SK model

Let us now apply the general formalism of the previous section to the SK model. The Weyl symbol of the SK Hamiltonian (20) is simply obtained by replacing the spin (angular momentum) operators by the classical spin variables and reads

$$H^w = -2 \sum_{i \neq j}^{N} J_{ij} s_i^z s_j^z - 2h \sum_{i=1}^{N} s_i^x \,, \tag{28}$$

with $J_{ij}$ the same random couplings as in Eq.(21). These spin variables evolve in time according to Eqs. (26). These equations have to supplemented with the initial conditions distributed according to the Wigner function. For simplicity we will consider simple product initial states $|\psi_0\rangle$, whose Wigner function $W(\{s_i^\alpha(0)\})$ also factorizes. Instead of the exact Wigner function we will choose its Gaussian approximation, where its first and the second moments are fixed by the mean and the variance of the corresponding quantum spin operators in the initial state:

$$\langle \psi_0 | \hat{s}_i^\alpha | \psi_0 \rangle = \overline{s_i^\alpha(0)} \,, \quad \frac{1}{2} \langle \psi_0 | \hat{s}_i^\alpha \hat{s}_i^\beta + \hat{s}_i^\beta \hat{s}_i^\alpha | \psi_0 \rangle = \overline{s_i^\alpha(0) s_i^\beta(0)} \,, \tag{29}$$

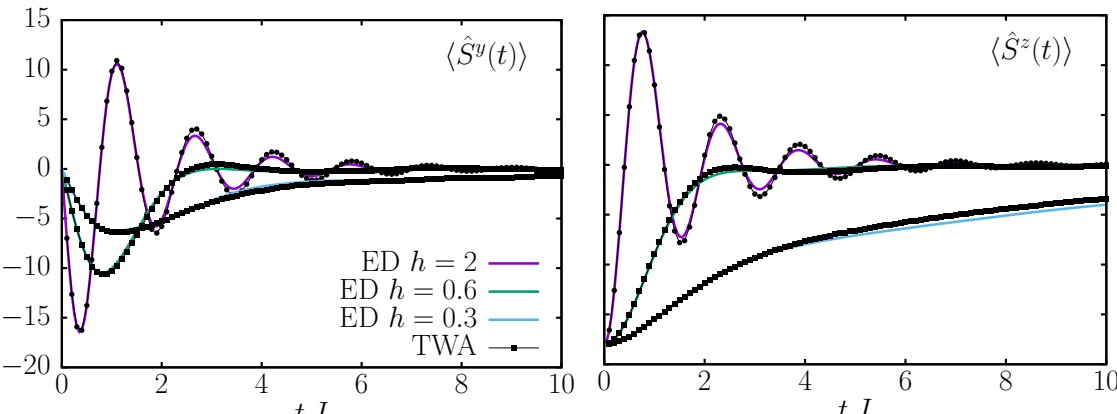

Figure 1: Comparison of TWA results to exact magnetization dynamics with SK couplings (21) for a fixed desorder realization at different transverse fields $h$ for $N = 18$ spins. The left (right) panels show the total magnetization along $y$ ($z$) direction: $\langle \hat{S}^y(t) \rangle$ ($\langle \hat{S}^z(t) \rangle$). Full lines ED simulations, dotted lines correspond to TWA simulations with $N_{samp} = 8000$.

for $\alpha, \beta = x, y, z$. As an example, the initial state $|\psi_0\rangle = |\downarrow\downarrow\ldots\downarrow\rangle$ corresponds to

$$\overline{\overline{s_i^z(0)}} = -1/2 \ , \ \overline{\overline{s_i^{x,y}(0)}} = 0 \ , \ \overline{\overline{s_i^\alpha(0)s_i^\beta(0)}} = \frac{1}{4}\delta_{\alpha\beta} \ .$$

One can show that this matching can be achieved for any product initial state [69]. The Gaussian Wigner function has the advantage that it is positive definite and easy to sample. Also, generally, the accuracy of the TWA is set by the second power of the effective Planck's constant, which is the same as the accuracy of the Gaussian approximation of the Wigner function [69]. Alternatively one can use a discrete Wigner function [66,67], which is also positive and which accurately describes all the moments of the spin operators in the initial state. We checked numerically that the results obtained using the Gaussian and the discrete Wigner functions do not have noticeable differences. We integrate numerically Eq.(26) and average at each time $t$ over $N_{samp}$ trajectories, whose initial conditions are distributed according to the initial Wigner function (29). For numerical integration, we use an adaptive fourth-order Runge-Kutta algorithm, fixing the error to $10^{-12}$.

Before analyzing the echo, let us consider the magnetization dynamics (19) with forward evolution, where we suddenly quench the system to the SK Hamiltonian (28). We check the validity of the TWA by comparing it with exact diagonalization (ED) [2]. In what follows, we focus for concreteness on the initial product state, where all the spins are polarized along the $z$-axis: $|\psi_0\rangle = |\downarrow\downarrow\ldots\downarrow\rangle$, but the validity of the method does not depend on this choice. In Fig. 1, we show results of the time evolution of the total spin components along the $y$ and $z$ axes for a fixed realization of the spin-spin couplings in the SK Hamiltonian. As expected, the TWA gives an excellent quantitative description of the forward time evolution of the magnetization for all simulated times and for different values of the transverse field $h$, covering both glassy and normal phases of the Hamiltonian. Furthermore, by increasing the system size $N$, TWA asymptotically approaches the exact quantum dynamics. This is shown in Fig.2, where we compare TWA to ED for fixed $h$ increasing $N$. In the inset of the same figure, we plot the absolute value of the difference between the two results Diff($\langle S_z(t) \rangle$), which clearly decreases with $N$.

---

[2] We address the exact quantum dynamics by employing the the method of Krylov sub-spaces in order to avoid full diagonalization, see e.g. Ref. [75].

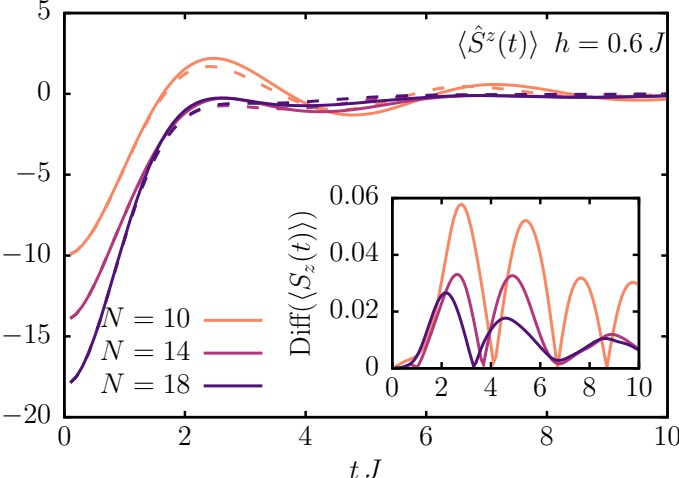

Figure 2: Comparison between ED (solid lines) and TWA (dashed lines) dynamics for $\langle \hat{S}^z(t) \rangle$ for the SK model (21) with a fixed disorder realization, fixed $h = 0.6 J$, and with different $N = 10$ , $14$ , $18$. In the inset we plot the absolute value of the difference between the two results. TWA simulations with $N_{samp} = 8000$.

As evident from the data, the TWA error reaches the maximum at an intermediate, system size-independent time before decreasing again at late times. The maximal (and the average) error diminishes with $N$. It is interesting that there is no clear signature of the Ehrenfest $t_{\mathrm{Ehr}}$ time in the forward evolution such that at sufficiently large $N$ the TWA correctly reproduces the magnetization dynamics at all times. This is to be contrasted with the echo dynamics, analyzed in the next section, where we will see that TWA breaks down after $t_{\mathrm{Ehr}}$, which for these parameters and largest analyzed $N = 20$ is given by $J t_{\mathrm{Ehr}} \approx 2$ (c.f. Fig. 5).

## 5.2 TWA for the short-range model

In the case of the short-range Hamiltonian, the TWA approach is the same as the one illustrated in the previous section, with the only difference of the short-range couplings as given by Eq.(22). In this case, $1/N(\sigma) \sim 1/\sigma$ acts an effective $\hbar$ and TWA is expected to fail at a time-scale set by $\sigma$, which is $N$-independent. In the short-range limit, for fixed finite $\sigma$, the semi-classical approximation does not reproduce the exact quantum dynamics in the thermodynamic limit. Indeed, in Fig.3(a.), we show the comparison of the TWA with the ED dynamics for $\langle \hat{S}_z(t) \rangle$ from the initial state $|\psi_0\rangle = |\downarrow \downarrow \ldots \downarrow\rangle$ at fixed $\sigma = 1$ varying $N = 10 \div 18$. The results might seem qualitatively in agreement with the exact dynamics. However, they do not improve with increasing the system size, as shown in the inset. When $N(\sigma) \sim \sigma$ is big enough, the TWA is accurate both at short and at long-times. In Fig.3(b.), we plot $\langle \hat{S}_z(t) \rangle$ at fixed $N = 18$ for different $\sigma = 1, 2, 6$. The difference between ED and TWA (displayed in the inset) shows how the reliability of the TWA grows by increasing $\sigma$.

# 6 Scrambling in the SK model

Let us now turn to the dynamics of the echo in the SK model and in its short-range version. In particular, we will study numerically the role of the number of spins $N$, the choice of the operator and of the range interactions for both observing the exponential growth of OTOC and for the validity of the semiclassical TWA approach. We first discuss the echo dynamics under the evolution of the all-to-all SK Hamiltonian given by Eq.(20).

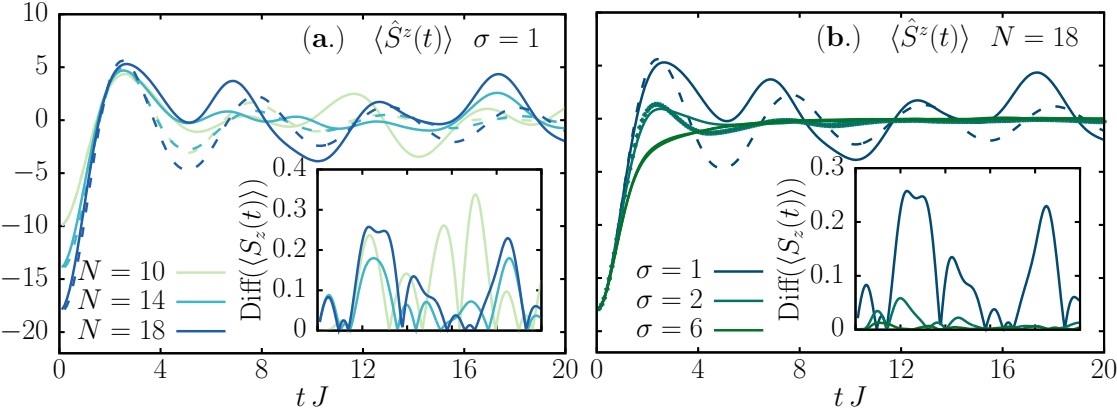

Figure 3: Comparison of ED (solid lines) and TWA (dashed lines) dynamics of $\langle \hat{S}^z(t) \rangle$ for a single realization of the short-range gaussian couplings (22) at $h = 0.6J$. TWA breakdown is set by $N(\sigma) \sim \sigma$, which is $N$-independent. (a.) Short-range couplings for fixed $\sigma = 1$ with different $N = 10, 14, 18$. In the inset, we plot the absolute value of the difference between the two results as a function of time. (b) Same as in a) but for fixed $N = 18$ and different range of interactions $\sigma = 1, 2, 6$. TWA simulations with $N_{samp} = 8000$.

Let us start by analyzing possible choices of the operators $\hat{A}$ and $\hat{B}$ according to the cases (i), (ii) and (iii) discussed in Sec. 3. We wish to compare the scaling with $N$ of the early and long-time behaviour of the echo in these three alternatives. For this definiteness, we focus on the magnetization along the $z$ axis and we consider (i) extensive-extensive $\hat{A} = \hat{B} = \hat{S}^z = \sum_j \hat{\sigma}_j^z$ [c.f. Eq. (19)], (ii) extensive-local $\hat{A} = \hat{S}^z$ with $\hat{B} = \sigma_i^z$ and (iii) local-local $\hat{A} = \hat{B} = \hat{\sigma}_i^z$, where the site $i$ is chosen randomly for each disorder realization. Notice that another possibility for (ii) discussed in Sec. 3 is $\hat{A} = \hat{\sigma}_j^z$ local with $\hat{B} = \hat{S}^z$ extensive. This choice in fact yields results identical to those of (i) with the $\langle A(t) \rangle$ and $\mu(t)$ simply scaled down by a factor of $N$. This follows from the fact that the expression for the echo (2) is linear in $\hat{A}$. We consider a fully polarized product initial state $|\psi_0\rangle = |\downarrow\downarrow \ldots \downarrow\rangle$, which automatically satisfies the requirement (16) and maximizes the difference between the initial and asymptotic value $\mathcal{A}(E) - \alpha_0 \sim -\alpha_0$ [c.f. Eq.(16)]. In fact, the energy of this fully polarized state lies in the middle of the spectrum of the Hamiltonian, therefore $\mathcal{A}(E) \sim 0$. This represents a generic choice suitable for studying the echo dynamics.

At early times the echo grows quadratically as predicted by Eq.(6), which in this case can be computed explicitly yielding (i) $\mu(t) = 8N h^2 t^2$ and (ii-iii) $\mu(t) = 8 h^2 t^2$. This perturbative expansion breaks down at $t^* \sim 1/\sqrt{J^2 + 4h^2}$. After $t^*$, $\mu(t)$ enters a non-perturbative regime, until it saturates at long-times to the value: (i) $\bar{\mu} \approx N^2$, (ii) $\bar{\mu} \approx N$, and (iii) $\bar{\mu} \approx 1$, as immediately follows from Eq. (9) for an infinite temperature state which has no magnetization correlations between different spins. This general behaviour is exemplified in Fig.4, where we show the exact quantum dynamics of the echo observable for (i-iii) for finite system sizes up $N = 8 \div 18$ for $h = 0.6J$, averaged over 50 desorder realizations. The figure further shows how the early time quadratic growth — red in the plot (iii) — breaks at a time, which is $N$-independent, the same is true for the collective observables. For (i-ii), the saturation value predicted by ETH is represented by dashed lines for each $N$ at the corresponding colour, displaying the existence of a parametric window that scales with $N$ that gives "room" for chaos to develop. On the other hand, the panel (iii) shows the saturation of the echo to one (green dashed line) leading the same dynamical behaviour of the echo, which is independent of $N$.

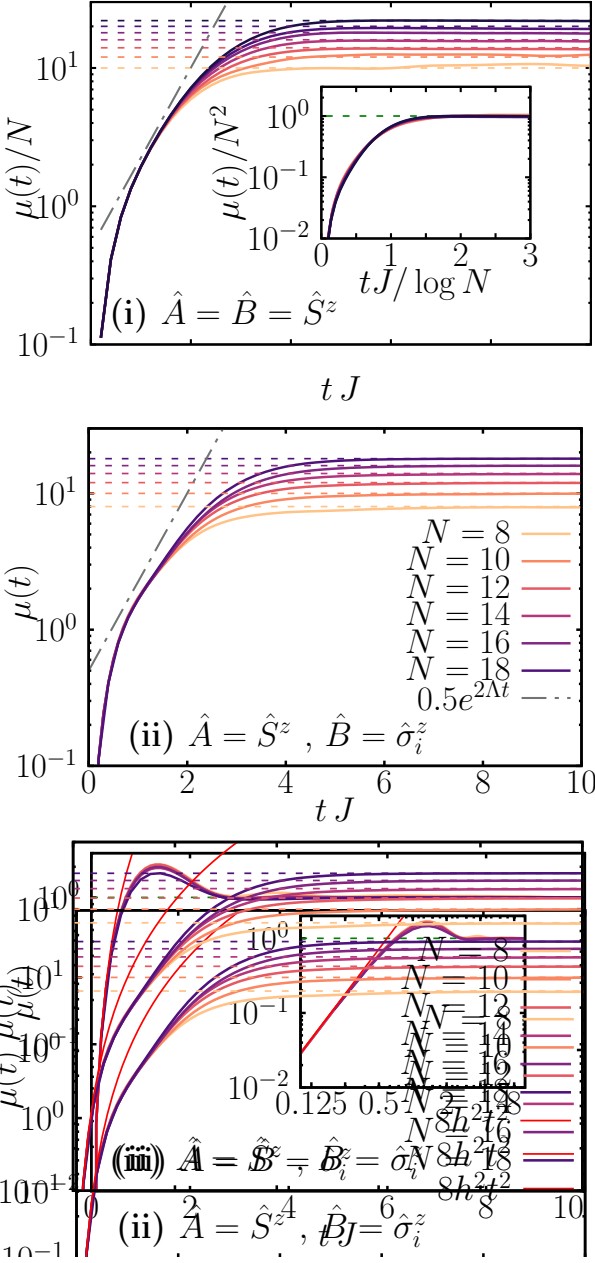

Figure 4: Exact scrambling dynamics $\mu(t)$ for different observables realizing three different scenarios (i-iii) discussed in Sec.3.3, for system sizes $N = 8 \div 18$. The saturation value as predicted by the ETH ansatz [cf. Eq.(9)] is illustrated by the dashed lines. To guide the reader's eyes, in (i-ii) we show an exponential function $f(t) = e^{2\Lambda t}/2$ in grey. The rate $2\Lambda = 1.5$ is extracted within TWA (see below). In (iii) the early time quadratic growth is plotted in red. (Top panel) (i) $\mu(t)/N$ for extensive-extensive operators $\hat{A} = \hat{B} = \hat{S}_z$. (Center panel) (ii) $\mu(t)$ for extensive-local operators $\hat{A} = \hat{S}^z$ with $\hat{B} = \sigma_i^z$. (Bottom panel) (iii) $\mu(t)$ for local-local $\hat{A} = \hat{B} = \hat{\sigma}_i^z$. In the inset of (i) we show $\mu(t)/N^2$ as a function of the rescaled time $tJ/\log N \sim tJ/t_{\mathrm{Ehr}}$ showing the long time scaling collapse of the echo for different values of $N$. (iii) The echo saturates to unity (green dashed line), while in the inset the same data are plotted in a doubly logarithmic scale. The plotted results correspond to a fully polarized initial state with $h = 0.6J$, averaged over 50 desorder realizations (see text for details).

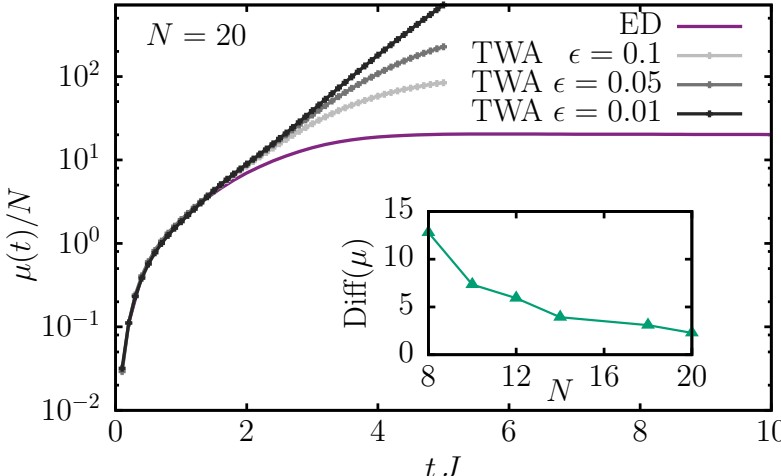

Figure 5: Comparison between the TWA scrambling dynamics $\mu(t)/N$ and the exact results at $N = 20$ varying $\epsilon$. An exponential fit of the TWA data for $\epsilon = 0.01$ with $f(x) = ae^{2\Lambda x}$ yields the exponent $2\Lambda = 1.5/J$. In the inset we show the difference between the TWA and ED results at fixed time $t = 2$ as a function of the system size $N$. At larger $N$, the quantum echo approaches the exponentially growing TWA prediction and then saturates. The results correspond to a fully polarized initial state with $h = 0.6J$ for a single disorder realizations. TWA with $N_{samp} = 20000$.

From this ED preliminary analysis for small system sizes, the echo observable already shows hints of exponential growth in the case of collective observables, see Fig.4 (i-ii). As evident from the data, this is possible due to the $N-$dependent saturation between the early-time and long-time behaviour.

Let us now focus on the case (i) for $\hat{A} = \hat{B} = \hat{S}^z$. By increasing $N$, the non-perturbative time-regime extends and the late time dynamics collapses if we plot $\mu(t)$ vs $tJ/\log(N)$, as shown in the inset of the same Fig.4 (i). This time-scale is compatible with the Ehrenfest time defined in Eq.(27), meaning that the echo has an asymptotic form $\mu(t) = N^2 f(tJ/t_{\text{Ehr}})$. Hence this scaling analysis shows that the intermediate, non-perturbative regime of exponential growth extends for $t^* < t < t_{\text{Ehr}}$, with the latter being divergent in the thermodynamic limit.

Since the quantum exponential growth is restricted in a time interval of width $\propto \log N$, a very slow function of its argument, one needs a numerical approach alternative to ED to simulate sufficiently large $N$ and fully appreciate the exponential growth numerically. In order to study the $\mu(t)$ dynamics before the Ehrenfest time, we resort to the TWA. As discussed in Sec.5.1, for this model the semi-classical approach correctly describes the expectation value of the observables in this time-regime. In Fig.5, we show TWA results in comparison with ED, for a single-disorder realization at finite size $N = 20$. After a short transient time, the TWA data exhibit a clear exponential growth, whose extent is determined by the parameter $\epsilon$, representing the strength of the perturbation, see Eq.(1). This situation is analogous to what happens in chaotic classical systems with compact phase-space. There, the ratio between the distance of two nearby trajectories, initially separated by $\epsilon$, ultimately saturates at a typical value fixed by the maximum available separation. For larger $\epsilon$ this saturation happens earlier, hence there is a shorter domain of exponential growth. The difference between exact ED and TWA data at fixed time, Diff($\mu$) diminishes with the system size as indicated in the inset of Fig.5. This result is consistent with the asymptotic accuracy of the TWA in the large $N$-limit, as discussed above for the magnetization. However, for long times, unlike for the magnetization, this difference

can be arbitrarily large as $\epsilon \to 0$.

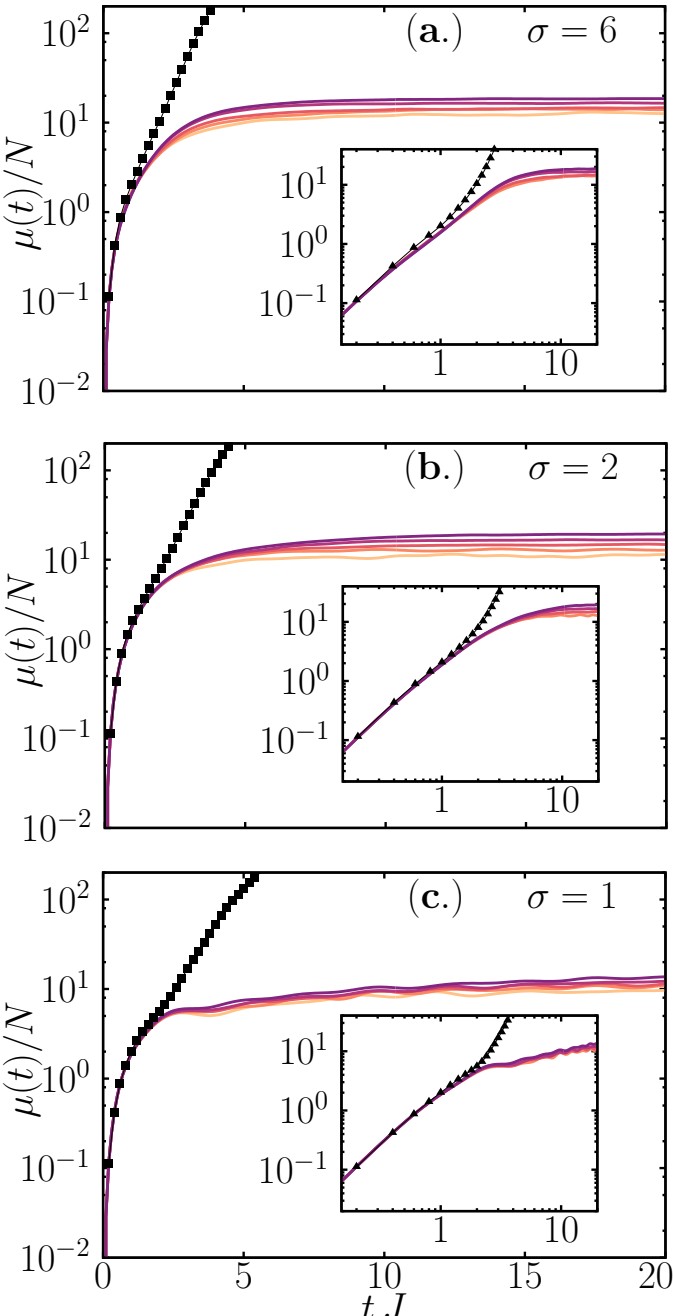

Figure 6: Absence of the exponential growth for the scrambling dynamics induced by short range couplings (22) for different $\sigma$. We compare exact quantum ED dynamics for different system sizes $N = 8 \div 16$ (increasing color intensity) with TWA results for $N = 16$ and $\delta = 10^{-2}$ (dotted black lines). Panels (a), (b), and (c) refer to decreasing interaction range $\sigma = 6, 2, 1$, respectively. The results correspond to a fully polarized initial state with $h = 0.6\,J$, averaged over 50 desorder realizations (see text for details).

Interestingly the TWA has an advantage over the ED method as it allows one to accurately extract the exponent characterizing the growth of the quantum echo in the thermodynamic

limit even using relatively small system sizes (see also Ref. [29] for the related discussion on the SYK model). In the case of the transverse field $h = 0.6J$ as in Fig. 5, an exponential fit yields $2\Lambda \sim 1.5/J$, while in general $\Lambda$ is an increasing function of $h$. The rate $\Lambda$ (sometimes referred to as the generalized Lyapunov exponent [76–78]) is related to the maximal Lyapunov exponent of the theory $\lambda_{max}$ [24,29]. The difference between the two comes from the different order of operations of taking logarithm and ensemble averaging. We would also like to point out that TWA is more accurate in extracting the Lyapunov exponent $\Lambda$ for the additional two reasons: 1) TWA does not know about the Ehrenfest time (a fully quantum time-scale) and its exponential growth lasts for many decades. 2) In TWA $\Lambda$ becomes independent on the system size even for relatively small $N$, allowing a precise estimate.

To summarize this discussion, TWA for the echo indeed breaks down at $t \propto \log N$, which are relatively short times unless $N$ is very big. But it breaks down in a smart way, which allows to predict quantum dynamics when $N$ becomes exponentially large. While this result seems to be paradoxical, it is correct and not incidental. By our arguments it should apply to any large N model, which has a diverging Ehrenfest time. This loosely follows from the fact that the main role of $N$ in dynamics is to set the value of $\hbar$, other corrections due to finite $N$ are small and very quickly disappear as $N$ becomes moderately large, of the order of 10. So the semiclassical-classical TWA dynamics effectively extrapolates $\hbar \to 0$ and is very efficient if we are interested in this limit.

Exactly the same considerations apply in the case of other observables, i.e. magnetization in the other directions $\hat{S}^x$, $\hat{S}^y$, see Appendix C for further examples.

## 6.1 Absence of exponential sensitivity in a short-range SK model

The exponential sensitivity of the echo disappears in the presence of local interactions. This happens simultaneously with the failure of the semi-classical TWA approximation. We consider the evolution of the same polarized initial state with the SK Hamiltonian with short-range Gaussian couplings (22). Short-range interactions result in at most a power-law growth of echo, in accordance with what was first observed in Ref. [24] and then proved in Ref [41]. In Fig. 6, we show the quantum ED evolution for a fixed system size at different values $\sigma = 1, 2, 6$ and compare them to the corresponding TWA results. As the plots show, for the short-range model $\sigma = 1, 2$ the echo growth according to the initial perturbative power until it crossovers to a slower polynomial growth best fitted by $\mu(t) \propto t^{0.5}$ consistent with Refs. [24,41] (see the inset) and the eventual saturation to the correct ETH value Eq. (15). As $\sigma$ increases one can observe a slow emergence of the non-perturbative intermediate time dynamics of fast echo growth, which is expected to crossover to the exponential growth in the limit $\sigma \to \infty$. From this plot it is also evident that the TWA fails after a shorter ($N$-independent) time, incorrectly showing the persistence of exponential growth of the echo even for the short-range model. These results can be re-phrased by saying that the effective Ehrenfest time becomes of the same order as the time of breakdown of the short time expansion, i.e. $t_{\text{Ehr}} \sim t^*$, leading to a lack of the semi-classical time-window necessary for the exponential quantum growth of the echo.

## 7 Discussion

In this work, we have studied the quantum echo dynamics and its exponential divergence in time in the Sherrington-Kirkpatrick model with transverse field. We have argued that, by choosing collective observables and an initial state such that the initial value of the observable is thermodynamically different than its stationary value, the echo grows exponentially, with the same rate of the underlying semi-classical theory. On the other hand, the presence of short-

range interactions results in the absence of exponential sensitivity in the quantum dynamics [24,52] as a result of the lack of a well defined semi-classical limit. In this case, understanding the nature of the non-perturbative polynomial regime remains an open question, beyond the scope of the present work.

Overall, we would like to emphasize that the echo (and the OTOC in general) can be used as a precise probe of failure of a classical analysis, exactly in the spirit of the seminal paper by Larkin and Ovchinnikov [16]. Indeed, the forward evolution of observables like the magnetization, is reproduced by the semi-classical evolution up to times which go well beyond $t_{\text{Ehr}}$ and can even extend all the way to infinity. Conversely, the semiclassical description of the OTOC breaks down precisely at $t_{\text{Ehr}}$ and it allows one to clearly identify the Ehrenfest time as the breakdown time of the classical evolution.

Because of the connection between the echo (the square commutator) and the expectation value (the variance) of the observables under effective time reversal, our findings are directly relevant to experiments allowing one to access exponential signatures of chaos in atomic experiments. A more general and open question concerns the full distribution of the echo operator. We observed numerically that higher cumulants of the echo signal produce deviations between the ED and the TWA predictions even before the Ehrenfest time. We will leave this analysis for future work.

## Acknowledgements

We acknowledge useful discussions with Rosario Fazio, Efim Rozembaum, Antonello Scardicchio, Markus Schmidt, Dries Sels, Xhek Turkeshi and Jonathan Wurtz. We thank the facilities of the Boston University Shared Computing Cluster, over which we run all the numerical simulations.

**Funding information** Research of A.P. was supported by NSF DMR-1813499 and AFOSR FA9550-16-1-0334. SP thanks Boston University's Condensed Matter Theory Visitors program for support. Part of this work has been carried out during the workshop "Breakdown Of Ergodicity In Isolated Quantum Systems" at the Galileo Galilei Institute (GGI) in Florence.

## A  Out-of-time ordered correlators in Bopp representation

In this Appendix, we first derive the semiclassical expression for the echo observable and the square-commutator [cf. Sec.2] using Bopp formalism and show that in the semi-classical limit both quantities contain the square of the derivatives of the classical trajectories with respect to the initial conditions. This implies that both the echo observable and the square commutator encode the classical Lyapunov exponent.

Let us start by introducing Bopp formalism. Wigner-Weyl quantization is intrinsically connected with symmetric Bopp representation of quantum operators [49]. This allows to map operators to functions of phase space variables without any need of performing tedious partial Fourier transforms. In particular, bosonic creation and annihilation operators in Bopp representation read

$$\hat{a}^{\dagger} \rightarrow \alpha^* - \frac{1}{2}\frac{\partial}{\partial \alpha}, \quad \hat{a} \rightarrow \alpha + \frac{1}{2}\frac{\partial}{\partial \alpha^*}. \tag{30}$$

Then, the Weyl symbol of, for example, the number operator is obtained by simply writing it

in Bopp representation

$$n^w = (\hat{a}^\dagger \hat{a})^w = \left( \alpha^* - \frac{1}{2} \frac{\partial}{\partial \alpha} \right) \alpha = \alpha^* \alpha - \frac{1}{2} \ .$$

Interestingly, Bopp formalism immediately allows one to compute non-equal correlation functions e.g.

$$\left( \hat{a}^\dagger(t_1) \hat{a}(t_2) \right)_w = \alpha^*(t_1)\alpha(t_2) - \frac{1}{2} \frac{\partial \alpha(t_2)}{\partial \alpha(t_1)} \ ,$$

where the derivative to respect to $\alpha(t_1)$ represents the non-equal time response. One can show that time ordered correlation functions always allow for a casual representation in the language of Bopp operators, while OTOC do not allow for such a representation [49, 51]. One can also write Bopp operators in a more compact form

$$\hat{a}^\dagger(t) \to \alpha^*(t) + \frac{i\hbar}{2}\{\alpha^*(t), \cdot\}, \quad \hat{a}(t) \to \alpha(t) + \frac{i\hbar}{2}\{\alpha(t), \cdot\} \ , \tag{31}$$

where $\{\cdot, \cdot\}$ stands for the classical Poisson bracket. In Bopp representation the creation and annihilation operators (and similarly the momentum and the coordinate operators) map to the corresponding phase space variables plus half of the Poisson bracket.

For more complicated operators, like non-linear bosonic variables or spin operators, this simple interpretation is lost as generally higher order derivatives emerge. In order to derive the semi-classical limit of OTOC at order $\hbar^2$, it is enough to keep at most the second-order expansion in $\hbar$ of the Bopp operator. In particular, for a generic time-dependent operator $\hat{B}(t)$, Bopp representation can be written as

$$\hat{B}(t) \to B_t + \hbar D_{B_t}^{(1)} + \hbar^2 D_{B_t}^{(2)} \ , \tag{32}$$

where $B_t$ is the Weyl symbol of the operator $\hat{B}$ evaluated at time $t$, the linear order is given by half of the Poisson brackets $D_{B_t}^{(1)} = i/2\{B_t, \cdot\}$, and $D_{B_t}^{(2)}$ contains the second-order derivatives and its explicit form depends on the operator $\hat{B}(t)$. For example, for spin operators as $\hat{B}(t) = \hat{\mathbf{S}}(t)$ this correspondence gives

$$B_t = \mathbf{S}_t \ , \quad D_{\mathbf{S}_t}^{(1)} = \frac{i}{2}\{\mathbf{S}_t, \cdot\} = -\frac{i}{2} \mathbf{S}_t \times \boldsymbol{\nabla} \ , \quad D_{\mathbf{S}_t}^{(2)} = -\frac{1}{8}\left[ \boldsymbol{\nabla}_t + (\mathbf{S}_t \cdot \boldsymbol{\nabla}_t)\boldsymbol{\nabla}_t - \frac{1}{2}\mathbf{S}_t \boldsymbol{\nabla}_t^2 \right] \ , \tag{33}$$

where $\boldsymbol{\nabla}_t = \partial/\partial \mathbf{S}_t$, i.e. see Ref. [49]. The second order contribution can be re-written in a more compact way as

$$D_{\mathbf{S}_t}^{(2)} = -\frac{1}{8} \frac{\partial}{\partial \mathbf{S}_t} + A_{\alpha\beta\gamma}^{\mathbf{S}} S_t^\alpha \frac{\partial^2}{\partial S_t^\beta \partial S_t^\gamma} \ , \tag{34}$$

where one has to sum over $\alpha, \beta, \gamma = x, y, z$ and the coefficients of $A_{\alpha\beta\gamma}^{\mathbf{S}}$ are determined explicitly from Eq.(33), e.g. $A_{xxx}^{S^z} = 0$, $A_{zzz}^{S^z} = 1/16$ or $A_{zxx}^{S^z} = -1/16$, etc.. In the case of operators which are linear in the creation and annihilation operators (or in the position and momentum operators) the second-order term vanishes $D^{(2)} = 0$ and one gets Eqs.(30-31).

These formulae can be used in constructing Weyl symbols for various time-dependent expectation values [49, 51] and, in particular, to compute out-of-time ordered correlators. To do so, we consider the Bopp representation of $\hat{B}(t)$ (32) and the corresponding one for $\hat{A}(0)$

$$\hat{A}(0) \to A_0 + \hbar D_{A_0}^{(1)} + \hbar^2 D_{A_0}^{(2)} \ .$$

To compute the semi-classical limit the echo discussed in Sec.2, we evaluate the Weyl symbol of several correlation functions, e.g.

$$\left(\hat{B}(t)\hat{A}\hat{B}(t)\right)_w = (B_t + D^{(1)}_{B_t} + D^{(2)}_{B_t})(A_0 + D^{(1)}_{A_0} + D^{(2)}_{A_0})B_t \,,$$

and we simplify the resulting expressions. After a tedious calculation, the Weyl symbol of the echo observable (2) reads

$$\left([\hat{B}(t),[\hat{B}(t),\hat{A}(0)]]\right)_w = \hbar^2\left[3\left(D^{(1)}_{B_t}\right)^2 A_0 - D^{(2)}_{B_t}B_tA_0 + D^{(2)}_{A_0}B_t^2 + A_0\,D^{(2)}_{B_t}B_t\right]\,, \qquad (35)$$

while for the square commutator $c(t) = -\langle[\hat{B}(t),\hat{A}(0)]]^2\rangle$ one finds

$$-\left([\hat{B}(t),\hat{A}(0)]]^2\right)_w = -4\hbar^2\left(D^{(1)}_{A_0}B_t\right)^2 = \hbar^2\{A_0,B_t\}^2 \,. \qquad (36)$$

In order to check Eqs.(35-36) let us consider as simple example $\hat{A}(0) = \hat{a}^2$ and $\hat{B}(t) = a^\dagger(t)$ and compute the equal time result at $t = 0$. One one side, the exact commutation relation for the bosonic operators immediately gives $-[\hat{a}^\dagger,\hat{a}^2]^2 = -4\hat{a}^2$ and $[\hat{a}^\dagger,[\hat{a}^\dagger,\hat{a}^2]] = 2$. On the other hand, it is straightforward to check that Eqs.(35-36) lead to $-([\hat{a}^\dagger,\hat{a}^2]^2)_w = -4\alpha^2$ and $([\hat{a}^\dagger,[\hat{a}^\dagger,\hat{a}^2]])_w = 2$. In fact, the Bopp representation (30) for $\hat{B} = \hat{a}^\dagger$ gives $B = \alpha^*$, $D^{(1)}_B = -\frac{1}{2\hbar}\partial/\partial\alpha$, $D^{(2)}_B = 0$, while for $\hat{A} = \hat{a}^2$ one has $A = \alpha^2$, $D^{(1)}_A = \alpha/\hbar\,\partial/\partial\alpha^*$, $D^{(2)}_A = \frac{1}{4\hbar^2}\partial^2/\partial\alpha^{*2}$.

It is well known that the classical limit of the square commutator (36) encodes the square of the derivatives of the classical trajectory to respect to the initial conditions [16–18]. This means that, whenever the classical limit is chaotic, $c(t)$ is expected to grow exponentially, with a rate given by twice the largest Lyapunov exponent. This can be directly seen also in the example discussed above with $\hat{A}(0) = \hat{a}^2(0)$ and $\hat{B}(t) = \hat{a}^\dagger(t)$, where Eq.(36) simply gives $c(t) \to -4\alpha^2(0)\left(\frac{\partial\alpha^*(t)}{\partial\alpha^*(0)}\right)^2$.

We now show that the same result applies to the semi-classical limit of the echo observable (35). This has been already discussed in Ref. [29], but, for the sake of completeness, we illustrate it here within our notations. Let us first analyze the previous simple example. Substituting the Bopp representation for $\hat{A}(0) = \hat{a}^2(0)$ and $\hat{B}(t) = \hat{a}^\dagger(t)$ into Eq.(35), and using the chain rule for the second-order derivatives, one gets

$$\left([\hat{a}^\dagger(t),[\hat{a}^\dagger(t),\hat{a}^2(0)]]\right)_w = \frac{1}{2}\left[3\left(\frac{\partial\alpha(0)}{\partial\alpha(t)}\right)^2 + \left(\frac{\partial\alpha^*(t)}{\partial\alpha^*(0)}\right)^2 + 3\alpha(0)\frac{\partial^2\alpha(0)}{\partial\alpha^2(t)} + \alpha^*(t)\frac{\partial^2\alpha^*(t)}{\partial\alpha^{*2}(0)}\right]\,,$$

which, exactly as the square commutator, is dominated by the square of the derivatives of the classical trajectory to respect to the initial conditions.

Let us now prove it for spin operators, which are the subject of the present work, whose Bopp operators are given by Eq.(33-34). We fix for definiteness $\hat{A}(0) = \hat{S}^z(0)$ and $\hat{B}(t) = \hat{S}^z(t)$, which has been considered in our numerical calculations, see e.g. Fig.4. Ignoring factors of the order of the unity and keeping only the second order derivatives in Eq.(34), a straightforward calculation yields

$$\left([\hat{S}^z(t),[\hat{S}^z(t),\hat{S}^z(0)]]\right)_w \sim \hbar^2 D^{(2)}_{S^z_0}S^{z\,2}_t \sim \hbar^2\left[\left(\frac{\partial S^z_t}{\partial S^\beta_0}\right)\left(\frac{\partial S^z_t}{\partial S^\gamma_0}\right) + S^z_t\frac{\partial^2 S^z_t}{\partial S^\beta_0\partial S^\gamma_0}\right]\,, \qquad (37)$$

where one should sum upon the indices $\beta\gamma = x, y, z$. In Eq. (37) we kept only the third term appearing in Eq.(35), as the calculation of the other terms is analogous. Eq.(37) shows that the semi-classical echo observable is proportional to the square of the derivatives of the classical spin trajectory $S^z_t$ to respect to the initial conditions $S^{x,y,z}_0$. Thus, exactly as the square-commutator, the semi-classical $\mu(t)$ encodes twice the Lyapunov exponent in presence of classical chaos.

# B  Derivation of the TWA as the saddle point of the path-integral formulation

In this section, we sketch the steps for the derivation of the TWA within the path integral formalism providing its formal justification in the large $N$ limit. Feynman's path integral representation of the time evolution is well known to connect quantum and classical dynamics [79]. As such, it provides a convenient framework which allows to define classical evolution as an appropriate saddle point and to find the leading quantum corrections. If one is interested in kinetic type approaches, it is convenient to work in the Schrödinger representation where one can develop diagrammatic expansions within the Keldysh path integral [80]. However, if the dynamics are far from equilibrium and the effective $\hbar$ is the only small parameter then it is convenient to work in the Heisenberg picture, where the density matrix only enters through the initial conditions. As we discussed in the main text, formally one can exactly map dynamics of spins into the dynamics of Schwinger bosons using Eqs. (23).

For simplicity we will focus here only on expectation values of time dependent observables. This analysis can be extended in a similar fashion to analyze various non-equal time correlation functions including OTOC [49,68]. Let us assume that our observable of interest is represented by some operator $\hat{O}$. Then in the Heisenberg representation its expectation value is given by

$$\langle \hat{O}(t) \rangle = \text{Tr}\left[ \hat{\rho}_0 \, T_K \, e^{\frac{i}{\hbar} \int_0^t \hat{H}(\tau)d\tau} \, \hat{O} e^{-i \int_0^t \hat{H}(\tau)d\tau} \right], \tag{38}$$

where $T_K$ denotes the time ordering along the Keldysh contour with later times appearing closer to the operator $\hat{O}$. The path integral representation for this expectation value is obtained by Trotterization of the time evolution operators and inserting resolution of identity through coherent states between each Trotter step. Details of the derivation of such path integral can be found in Refs. [49,68]; here we only quote the final result:

$$
\langle \hat{O}(t) \rangle = \int d\boldsymbol{\alpha}_0 d\boldsymbol{\alpha}_0^* \, W(\boldsymbol{\alpha}_0, \boldsymbol{\alpha}_0^*) \int \mathcal{D}\boldsymbol{\alpha}\mathcal{D}\boldsymbol{\alpha}^* \mathcal{D}\boldsymbol{\eta}\mathcal{D}\boldsymbol{\eta}^* \, O^w(\boldsymbol{\alpha}(t), \boldsymbol{\alpha}(t)^*)
$$
$$
\exp\Big\{ \int_0^t d\tau \Big[ \eta_j^*(\tau)\frac{\partial \alpha_j(\tau)}{\partial \tau} - \eta_j(\tau)\frac{\partial \alpha_j^*(\tau)}{\partial \tau} + iH^w\Big(\boldsymbol{\alpha}(\tau) + \frac{\boldsymbol{\eta}(\tau)}{2}, \boldsymbol{\alpha}^*(\tau) + \frac{\boldsymbol{\eta}^*(\tau)}{2}, \tau\Big)
$$
$$
- iH^w\Big(\boldsymbol{\alpha}(\tau) - \frac{\boldsymbol{\eta}(\tau)}{2}, \boldsymbol{\alpha}^*(\tau) - \frac{\boldsymbol{\eta}^*(\tau)}{2}, \tau\Big) \Big] \Big\}, \tag{39}
$$

where $\boldsymbol{\alpha} \equiv \{\alpha_j\}$, $\boldsymbol{\alpha}^*$, $\boldsymbol{\eta}$, $\boldsymbol{\eta}^*$ are the classical (symmetric) and quantum (antisymmetric) bosonic fields with the index $j$ running over both different sites and different Schwinger boson flavors. The vectors $\boldsymbol{\alpha}_0$ and $\boldsymbol{\alpha}_0^*$ represent initial "classical" fields, which are distributed according to the Wigner function. We highlight that in this form the path integral representation of the evolution is exact and both the Weyl symbols of the Hamiltonian and the observable and the Wigner function automatically emerge. The TWA emerges from the path integral by taking the saddle point approximation of the action (integrand) with respect to quantum variables $\eta_j(\tau)$ and $\eta_j^*(\tau)$. It is easy to see that this saddle point approximation is equivalent to linearizing the difference between Hamiltonians $H^w$ on the forward and the backward path to the linear order in $\boldsymbol{\eta}$:

$$
H^w\Big(\boldsymbol{\alpha}(\tau) + \frac{\boldsymbol{\eta}(\tau)}{2}, \boldsymbol{\alpha}^*(\tau) + \frac{\boldsymbol{\eta}(\tau)^*}{2}\Big) - H^w\Big(\boldsymbol{\alpha}(\tau) - \frac{\boldsymbol{\eta}(\tau)}{2}, \boldsymbol{\alpha}^*(\tau) - \frac{\boldsymbol{\eta}^*(\tau)}{2}\Big)
$$
$$
= \eta_j(\tau)\frac{\partial H^w(\boldsymbol{\alpha}(\tau), \alpha^*(\tau))}{\partial \alpha_j(\tau)} + \eta_j^*(\tau)\frac{\partial H^w(\boldsymbol{\alpha}(\tau), \boldsymbol{\alpha}^*(\tau))}{\partial \alpha_j^*(\tau)} + \mathcal{O}(|\eta^3(\tau)|). \tag{40}
$$

By integrating out over quantum $\boldsymbol{\eta}(\tau)$ and $\boldsymbol{\eta}^*(\tau)$ variables, one enforces the deterministic evolution of the classical variables $\boldsymbol{\alpha}(\tau)$ and $\boldsymbol{\alpha}^*(\tau)$ according to the standard Hamiltonian equations of motion (25)

$$i\frac{d\alpha_j}{dt} = \frac{\partial H^w(\boldsymbol{\alpha}(\tau), \boldsymbol{\alpha}^*(\tau))}{d\alpha_j^*}(\tau) \equiv \{\alpha_j(\tau), H^w(\boldsymbol{\alpha}(\tau), \boldsymbol{\alpha}^*(\tau))\} .$$

As discussed in the main text, these equations are equivalent to the Hamiltonian equations for spins (angular momentum) variables if one goes back from complex $\boldsymbol{\alpha}$ variables to standard classical angular momentum variables [49]. If one ignores fluctuations in the initial conditions setting $\boldsymbol{\alpha}_0$ to a fixed mean field value, and interprets the Schwinger boson components for each spin $\alpha_0$ and $\alpha_1$ as the components of the wave function, then TWA reduces to the Dirac's variational principle. Let us note, however, that one needs much stronger assumptions about the nature of initial state and absence of unstable chaotic dynamics in order to justify this variational principle. In most cases it leads to very poor predictions for the dynamics even if the effective $\hbar$ controlling the saddle point approximation is very small. Conversely, TWA is not relying on the assumptions about the initial state.

As a final ingredient for justifying the TWA for the SK model, we need to show that $1/N$ plays the role of the effective Planck's constant. This can be readily seen by analyzing the effect of neglected in cubic terms in $\eta$ of Eq. (40) on the observable $\langle \hat{O}(t) \rangle$. Let us show that these terms indeed are suppressed by $1/N$. We compute the derivatives of the Weyl symbol of the Hamiltonian (20). Ignoring numerical prefactors of the order of unity, the neglected terms in the path integral are of the type

$$\frac{J}{\sqrt{N}} \sum_{ij} g_{ij}\, \alpha_i^*(\tau)\eta_i(\tau)\eta_j^*(\tau)\eta_j(\tau) + c.c.,$$

where $g_{ij}$ are the Gaussian random variables with zero average and unit variance, appearing in the couplings (21). Here for simplicity, we suppress a spin Schwinger boson index in $\boldsymbol{\alpha}, \boldsymbol{\eta}$ variables since it is unimportant for the scaling and we only keep the site index. In Ref. [49] it was shown that these terms result in the cubic response of the observable $O^w$ to the infinitesimal quantum jumps on the classical $\boldsymbol{\alpha}$ fields integrated over time:

$$\delta O(t) \sim \frac{J}{\sqrt{N}} \int_0^t d\tau \int d\boldsymbol{\alpha}_0 d\boldsymbol{\alpha}_0^* W(\boldsymbol{\alpha}_0, \boldsymbol{\alpha}_0^*) \sum_{ij} g_{ij} \left( \alpha_i^*(\tau) \frac{\partial}{\partial \alpha_i^*(\tau)} \frac{\partial}{\partial \alpha_j^*(\tau)} \frac{\partial}{\partial \alpha_j(\tau)} + c.c. \right)$$
$$O^w(\boldsymbol{\alpha}(t), \boldsymbol{\alpha}^*(t)) . \tag{41}$$

To simplify the further discussion suppose that $O^w$ is linear in spin variables, say it represents the magnetization as analyzed in the main text $O^w = s_k^z \sim \alpha_k^* \alpha_k$. One can see that the deviation of the expectation value of the observable from its TWA value is suppressed by at least $1/N$ factor as

$$\delta O(t) \sim \frac{O^{TWA}}{N} . \tag{42}$$

The first $1/\sqrt{N}$ comes from the coupling's normalization in Eq.(41), and the other contributions come from the double summation. Anyhow, only terms with $ij \neq k$ should be accounted, for which it is easy to see that each derivative contributes with

$$\frac{\partial}{\partial \alpha_j} s_k \sim J \frac{g_{jk}}{\sqrt{N}} s_k .$$

This observation immediately follows from the structure of the classical equations of motion as this derivative represents the response of the $k$-th spin to an infinitesimal perturbation of

the $j$-th spin, which is suppressed (at least at short times) by the coupling constant, which scales as $1/\sqrt{N}$. Combining all the factors of $N$ and performing the disorder average we get immediately the estimate in Eq.(42).

We note that there is a standard issue of controllability of TWA (as well as of any other numerical method) at long times, which is very difficult to resolve analytically. In the present work, we show that for the echo, or OTOC, the TWA works until the Ehrenfest time, which scales as $\log(N)$, while for standard forward observables the mistake remains suppressed at all times.

## C  Echo dynamics for the magnetization along $y$

In Sec.3, we argued that the choice of the initial state to respect to the observable is crucial in order to ensure space for chaos to develop and in Sec.6 we showed the results of its exponential growth for $\hat{S}_z$. Below, we show the same analysis for the equivalent operators $\hat{S}^x$, $\hat{S}^y$.

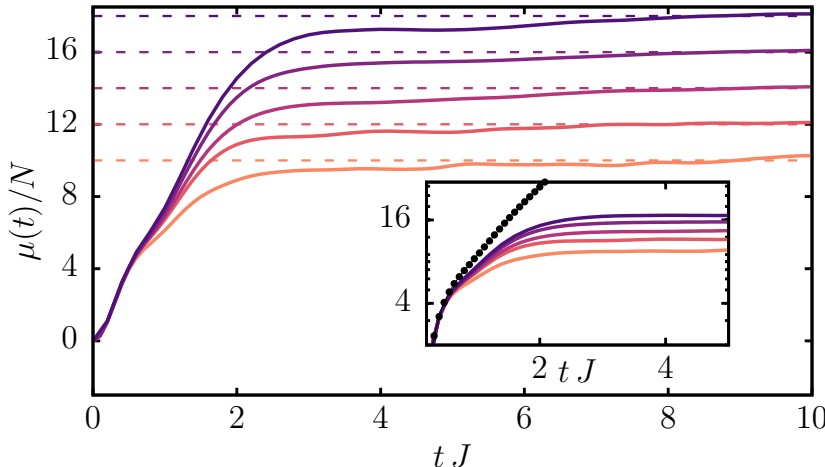

Figure 7: Exact scrambling dynamics $\mu(t)/N$ with the SK hamiltonian (20) for $\hat{A} = \hat{B} = S^y$ for $N = 10 \div 18$ increasing color's intensity. Dashed in the plot the ETH saturation value for finite $N$. In the inset the date are plotted in a semi-log scale to display the exponential growth before saturation. An exponential fit of the TWA data with $f(x) = a e^{2\Lambda x}$ yields the exponent $2\Lambda \simeq 1.1/J$. The results correspond to a fully polarized initial state in the $y$ direction with $h = 0.1J$, for 50 desorder realizations.

Let us first consider $\hat{A} = \hat{B} = \hat{S}^y$ with the initial state $|\phi_0\rangle = |LL\ldots L\rangle$ fully polarized in the $y$ direction ($\hat{\sigma}_i^y |L\rangle_i = -|L\rangle_i$). As for the $z$ direction, also the energy $E$ of $|\phi_0\rangle$ lies the middle of the spectrum, hence the magnetization along the $y$ always vanishes at long-times, i.e. in Eq.(16) the difference $\mathcal{S}^y(E) - S_0^y \simeq -S_0^y$ is maximized. Therefore the same conclusions of Sec.6 for $\hat{S}^z$ hold in this case. The resulting behaviour is exemplified in Fig.7, where we show the exact quantum dynamics of the echo observable at finite system size up $N = 8 \div 16$ for $h = 0.1$, averaged over 50 desorder realizations.

On the other hand, for $\hat{A} = \hat{B} = \hat{S}^x$ with $|\chi_0\rangle = |++\cdots+\rangle$ ($\hat{\sigma}_i^x |+\rangle_i = |+\rangle_i$) the situation slightly changes. In this case, the Hamiltonian (20) has a transverse field in the $x$-direction, and $\langle \hat{S}^x(t) \rangle$ attains a non-vanishing value at long-times, which changes the stationary value of the echo (9). With this choice of the initial state, we have $\alpha_0 = S_0^x = N$. Anyhow, by choosing a small transverse field, the dynamics is such that $\mathcal{S}^x(E)$ is still finite (but small) and the difference $\mathcal{S}^x(E) - \alpha_0$ not only is extensive, but big enough to appreciate the exponential

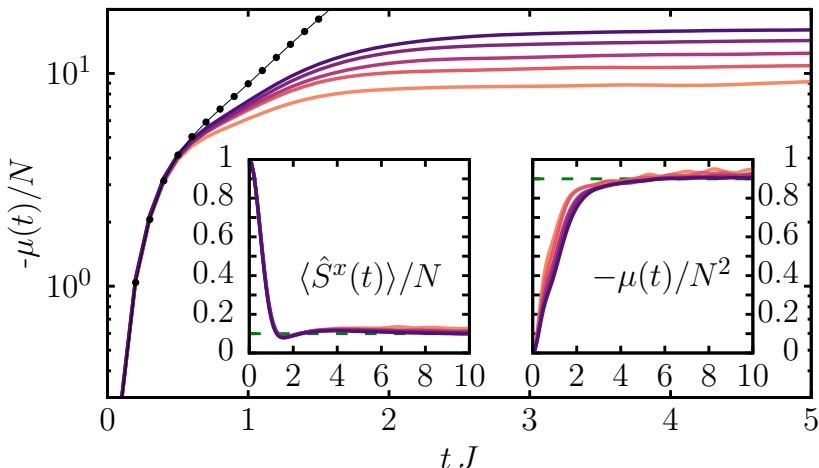

Figure 8: Echo dynamics $-\mu(t)/N$ of the operator $\hat{A} = \hat{B} = \hat{S}^x$. We compare exact quantum ED dynamics for different system sizes $N = 10 \div 18$ (solid lines with increasing color intensity) with TWA results for $N = 16$ (dotted black). An exponential fit of the TWA data with $f(x) = ae^{2\Lambda x}$ yields the exponent $2\Lambda \simeq 1.3/J$. In the left inset we show how the magnetization's dynamics $\langle \hat{S}^x(t) \rangle$ saturates to a finite value $\overline{S^x} \sim 0.1N$, while on the right the echo saturates to $\overline{\mu} \sim N(\overline{S^x} - \alpha_0) \sim 0.9N^2$. Dynamics from the fully polarized state along $x$, with $h = 0.2J$ over 50 desordor realizations, TWA with $N_{samp} = 20000$ and $\delta = 0.01$.

growth with exact numerics for small system sizes. The echo dynamics is displayed in Fig.8, where we show the exact quantum dynamics of the echo observable at finite system size up $N = 8 \div 18$ for $h = 0.2$, averaged over 50 desorder realizations. By increasing $N$, the ED results approach the TWA semi-classical dynamics, characterized by the exponent $2\Lambda \simeq 1.3$. In the insets we show how the saturation value $\overline{S^x} \simeq \mathcal{S}^x(E)$ affects the echo's saturation, leading to $\overline{\mu} \sim N(\mathcal{S}^x(E) - \alpha_0)$.

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
