# Peer review of "Quantum echo dynamics in the Sherrington-Kirkpatrick model"

_SciPost Physics, doi:SciPost Phys. 9, 021 (2020)_

## Round 1 · Referee Report · Anonymous (Referee 1) · 2019-11-29

Strengths

The authors address timely a subject of interest to a broad range of communities

Weaknesses

The article is not very clear, and makes some far fetched statements.

Report

The manuscript addresses a subject which is of current interest in a wide range of communities. The authors study the evolution of an echo defined for observables. This quantity is related to the original proposals of the Loschmidt echo which tried to characterize quantum chaos and irreversibility in quantum systems by the decay of the overlap between a state evolved with a Hamiltonian and another state evolved with (almost) the same (slightly perturbed) Hamiltonian. The subject has developed a lot in the last few years but using a different approach, which is characterizing quantum chaos by how information is scrambled over time. This is quantified by the (exponential) growth (or decay, depending on how it is defined) of the out-of-time ordered correlators. There has been an increase of interest since the deduction of an upper bound (essentially temperature) for the growth rate in certain maximally chaotic systems, that could possibly be related to black hole dynamics.

In such a context the authors provide some results which could be worthy of interest, but which in my opinion suffer from some problems.

I’ll first point out what I find very interesting. The authors point out the known difficulty in many body systems to find the alleged exponential growth of the OTOC. The authors make an interesting analysis showing that the problem is essentially due to two things: the type of initial state, and the type of observable. In order to have a large enough window to accommodate for an exponential growth the initial state should be extensively far from the saturation value, and at least one of the observables should be non-local.

These results are analytical for the quantity $\mu$ of Eq (2) which they define as echo. It does not depend on the model.

After the analytical results the authors proceed to show numerical results for the Sherrington-Kirkpatrick model, which is a system of spins with infinite range interaction that has a glass transition. In the numerical results the authors show first that the exact dynamics of $\mu(t)$ do indeed behave as predicted and then they use the semiclassical Truncated Wigner approximation (TWA) to try to show that they can extract the exponential behavior from the model (in the suitable initial state +observable combination) and in addition they show that this is not possible for limited range interaction.

The results (though not ground breaking) could be interesting enough to be published in light of the present interest of the subject in one and many-body complex systems.

However there are many concerns which I enumerate below which should be addressed before it can be accepted.

(1) I think the authors try to force the relation between $\mu$ and OTOC. Indeed the square commutator — at order $epsilon^2$ — appears in the second moment of $\mu$ but that is no reason to speak interchangeably of the behavior of one quantity and the other, especially of exponential growths (or decays, or whatever). I think this things should be clearly stated somewhere, or everywhere. If there is indeed a one-to-one relation to the growth of the OTOC, please show it. I’m not saying $\mu$ is not a good quantity to characterize scrambling, it is just not the most widely used, if it is indeed one, then prove it. I think for completeness a better explanation could be included in the manuscript.

(2) The explanation and justification of the TWA is rather scarce, I was not familiar with it and found it rather difficult to follow in the text. Since the subject appeals to a broad audience, then effort could (maybe should) be made so that the article is as self contained as possible.

(3) I think I missed the point of using the TWA altogether. Then again it might be me, but it could have been better explained. In Fig 5, if no TWA is plot, I wouldn’t bet on an exponential growth “window”. The TWA does indeed show an exponential growth, but after the Ehrenfest time. So, paradoxically, the exponential regime through the semiclassical approximation could be obtained when the validity of the approximation breaks (?).

These are my major concerns which prevent me from recommending publication. I have some other minor comments.

-In the introduction, the reference to Peres in the question of irreversibility [1-4] is correct, however when giving credit about Loschmidt echo, i guess the Jalabert & Pastawski (PRL 86,2490 (2001)) should be cited, and maybe also due to its impact Jacquod, Silvestrov & Beenakker (PRE, 64, 055203􏷖R􏷓).

-In page 3 the list (Refs 27-31) of works where the Lyapunov exponent has been found (or exponential sensitivity) seems incomplete.

-Im not sure about something in Eq. (1). They say $\hat{A}$ evolves first with $\hat{H}$, then a rapid rotation is applied ( $\epsilon\hat{B}$) and then it evolves with the reversed Hamiltonian $-\hat{H}$. The first forward evolution of $\hat{A}$ would be
\[
\hat{A}(t)=e^{-i\hat{H}t}\hat{A}e^{i\hat{H}t}
\]
but in Eq(1) the signs are not like this.

-Saying Eq (2) “contains OTOC” is far fetched.

-Page 10, The “TWA is known to asymptotically describe quantum echoes at short times”, please provide a reference.

-The “TWA for the SK model” section is not very clear.

-Section 6: “Let us now turn to the dynamics of the echo and hence OTOC…” This direct relation is not established clearly.

Requested changes

(stated in the report)

---

## Round 1 · Referee Report · Rodolfo Jalabert (Referee 2) · 2019-12-5

Report

Report on the manuscript “Quantum echo dynamics in the Sherrington-Kirkpatrick model”
by S. Pappalardi, A. Polkovnikov, and A. Silva

The present manuscript concerns the echo dynamics of quantum operators for the Sherrington-Kirkpatrick spin model with transverse magnetic field. Numerical and analytical approaches are used to investigate the conditions for the existence of an exponential growth of the echo observables. The subjects of quantum echoes and out-of-order correlators (OTOCs) treated in this manuscript are timely and the results obtained are interesting within the present theoretical efforts to understand the scrambling of quantum information.

A number of questions need to be addressed in order to assess the significance of the proposed contribution.

1) While the second moment of the echo operator $A_{\epsilon}(t)-A$ corresponds in lowest order in $\epsilon$ to the square commutator (or OTOC), the studied quantity $\mu(t)$ differs, in general, from the OTOC. Even in the case in which the initial state is an eigenstate of the operator $A$, the quantity $\mu(t)$ differs from the OTOC. It is true that in this case both quantities have as a contribution the maximally crossed term $AB(t)AB(t)$. However, in the general case, the term arising from the matrix element of $B(t)A^{2}B(t)$ is present in the double commutator, but not in $\mu(t)$. It is only for unitary operators that the maximally crossed term suffices to describe the OTOC. Therefore, the case of the local operators treated in this work would be comprised, since the on-site spin operators are at the same time Hermitian and unitary, but the non-local collective observables discussed in this work are not unitary.
The difference between the OTOC and $\mu(t)$ is not unimportant since the terms that make up for their difference have been shown to be crucial in obtaining the exponential growth of the former as the difference between parametrically larger components.
While the authors expressed that the properties of the distribution of the echo operator are left out for future work, it should be clearly indicated in this manuscript that $\mu(t)$ is a different object than the OTOC. In addition, the authors should try to establish a connection with previous results for $\mu(t)$ in the case that they exist in spin chains or in other kind of systems.

2) The possibility of observing an exponential growth of $\mu(t)$ is linked to the existence of a sufficiently long time-window between the initial (perturbative) power-law growth and the long-time saturation. And such a condition is shown to put constraints on the nature of the initial state and of the observable. This line of thought is supported by the numerical simulations and the implementation of the truncated Wigner approximation on the Sherrington-Kirkpatrick model for the chosen parameters. However, in a general case, no argument is given on why in the intermediate time-window the growth of $\mu$ should be necessarily exponential. This is a central issue of current interest since the behavior of the OTOC in the non-perturbative intermediate regime, as well as for long-times, are not simply dictated by the integrable or non-integrable character of the system, especially in the case where the classical analog is not well-defined.

3) The large N-limit is presented as a semi-classical one, determining one of the three requirements needed to obtain the exponential divergence of the echo observables. In addition, the large-N limit is linked with the improvement of the truncated Wigner approximation. However, the proper semi-classical limit of a spin system is that in which the number of possible spin components goes to infinity, while in the present work the spin 1/2 is kept throughout. The necessity of having a large N appears as a condition for enlarging the intermediate time-window, and it is thus unrelated to a semi-classical approximation. While, in interacting many-body systems, 1/N sometimes plays an analogous role than that of $\hbar$ for a one-body system, the usage of semiclassical limit as synonymous of large N is not justified in the present model.

4) The chosen model, in its long-range version, shares with the Sachdev-Ye-Kitaev model the feature of having a non-local (global) interaction, and both of them exhibit the exponential growth of the echo observables. It would be important to signal the relevant differences between the two models and why a non-local interaction seems to be a crucial ingredient in obtaining the above-discussed behavior.

5) A cleaner presentation of the manuscript is needed. Some of the figure captions refer to “bottom” and “top” panels while describing horizontal layouts (i.e; Fig.1), while the symmetrical confusion also appears (i.e. Fig.4). Often the ordinate axis-label is missing. Misspellings and repeated words are to be corrected.

Addressing the above-presented questions and deepening the discussion of the physics behind the numerical findings should increase significance of this interesting manuscript.

  • validity: -
  • significance: -
  • originality: -
  • clarity: -
  • formatting: -
  • grammar: -

Author:  Silvia Pappalardi  on 2020-02-17  [id 739]

(in reply to Report 2 by Rodolfo Jalabert on 2019-12-05)
Category:
answer to question
reply to objection
validation or rederivation

Question (1):

We thank the referee for this comment. Since a similar issue has been raised also by the Referee 1, we have added few lines in the introduction to the new version of the manuscript, improved the discussion in Section 2 and included a new small appendix with the details of the classical limit of the echo observables.

First of all, let us specify the general meaning that one can associate to OTOC. Even if in recent literature the name OTOC is usually related to correlators as $\langle \hat A \hat B(t) \hat A \hat B(t) \rangle$, ``out of time-ordered correlators'' OTOC can be referred to all the time-dependent correlation functions which have an unusual time-ordering. Namely, they describe correlation functions which cannot be computed with conventional techniques that assume casual time evolution and therefore need to be evaluated via extended Keldysh contours, like the one developed in [Aleiner, Faoro, Ioffe, Annals of Physics 2016]. In this sense, the echo observable $\mu(t)$ can be refereed as OTOC, since it contains terms as $\langle \hat B(t) \hat A \hat B(t) \rangle$, see Eq.(4).

In the second place, in the semi-classical limit, every kind of OTOC contains quadratic terms in the derivatives of the initial conditions, as was already known by Larkin and Ovchinnikov. This is true also for the echo $\mu(t)$. This statement can be shown by the use of time-dependent Bopp operators, a standard formalism of phase-space methods which allows computing time-dependent correlation functions, by representing quantum operators as phase-space variables. This is extensively discussed in the Appendix of [Schmitt, Sels, Kehrein and Polkovnikov, PRB 2019] for a semi-classical limit of the SYK. Anyhow, to be as self-consistent as possible, we have added a new Appendix to the manuscript, with the derivation of the semi-classical limit of $\mu(t)$, based on Bopp operators. We show for bosonic and spin operators, that the semi-classical echo observable contains quadratic terms in the derivatives of the classical trajectory to respect to the initial conditions, hence it encodes the Lyapunov exponent, exactly as the square commutator. For further details on this derivation, we refer to the response (1) to the Report 1, where we have reported the main steps of the proof.

In conclusion, we acknowledge that the square commutator $c(t) = - \langle [\hat B(t), \hat A]^2 \rangle$ contains a term $\langle B(t)A^2 B(t)\rangle$, which is missed by $\mu(t)$. For this reason, we added a sentence in Section 2. Since we find this very interesting, we would like to ask if the Referee could kindly point us the reference where ``the terms that make up for their difference have been shown to be crucial in obtaining the exponential growth of the former as the difference between parametrically larger components.''

Question (2):

We thank the referee for this comment since this point is indeed a non-trivial statement and it is exactly one of the purposes of the paper. If we believe that there are no other initial time scales, our claim can be re-stated as follows: If there exist a time-interval $t^\ast\leq t \leq t_{\text{Ehr}}$, where $t_{\text{Ehr}}$ diverges in the thermodynamic limit, then in the same interval the $\mu(t)$ should be semi-classical. Therefore it behaves like the square of the derivatives to respect to the initial conditions, see also the Refs.[16-18] of the manuscript and the new Appendix A. When the latter grows exponentially with a rate given by the classical (generalized) Lyapunov exponent, then - in that time interval - also the quantum $\mu(t)$ should grow exponentially with the same rate. On the other hand, when the system does not have a semi-classical analogue, i.e. $t_{\text{Ehr}}\sim t^*$ does not depend on the system size, there is no time-regime where the quantum $\mu(t)$ would behave semi-classically. Therefore there is no reason in principle to have exponential growth. This is typically the case of short-range interacting lattice systems, which do not have a classical analogue and for which it has been proved that the square-commutator grows at most exponentially fast [Kukulian et al.]. This statement is firmly supported by our numerics on long-range (semi-classical) and short-range (fully-quantum) models.

Question (3):

We thank the Referee for this comment since our previous version did not provide the underlying argument to our statements. We have added a discussion and a new appendix in order to better justify our claims. The justification can be summarized as follows. TWA can be proved as the saddle point of the Keldysh action in a path-integral formulation of the time-evolutions. Within this framework, representation of initial conditions through the Wigner function is exact, while the approximation occurs at the level of the time-evolution, which is solved by the saddle-point approximation expanding in small quantum fluctuations. Even if the present work deals with spins $1/2$, this all-to-all interacting model has a mean-field character and the large $N$-limit ensures the validity of the saddle point approximation, as confirmed by the numerics. More formally (see Appendix B), one can estimate the corrections to the saddle-point solution, which are found to scale as $1/N$.

Question (4):

We thank the referee for highlighting this point, since the presence of non-local interactions, shared by the SYK and SK models, is indeed crucial to have a semi-classical limit and hence to observe the exponential growth of the OTOC. Even if in the previous version we already pointed out that the models share several analogies, we have now added a sentence to signal that the SYK and SK differ in many respects. For example, they have different equilibrium phase-diagrams, which inevitably lead to differences in the dynamics. This is particularly evident at low temperatures, where the SK model possesses a glassy phase below a critical transverse field, i.e. see [Andreanov and Muller, PRL 2012]. On the other hand, such a phase-transition is absent in the SYK model. Anyhow, as pointed out by the Referee, the two models share the feature of having non-local all-to-all random interaction. Indeed, a semi-classical analysis was performed by one of us on the SYK using the TWA on fermionic bilinears, see [Schmitt, Sels, Kehrein and Polkovnikov, PRB 2019], where the authors found similar results concerning the exponential growth of the echo. In that paper, the role of the short-range interactions was not addressed, and the analysis presented in the present work shows how the non-locality of the interactions affects the classical limit (even on a system of spins $1/2$) and therefore the exponential growth. Also in that paper, there was no analysis of the observables and the echo operator and the choice, which satisfies the criteria we derived here was rather coincidental. We believe that finding these conditions is another important new result of the present work.

Question (5):

We changed the figure's captions and added the ordinate axis where missing. We made an effort to improve the presentation, correcting misspellings and rewriting some sections.

Summary of changes: - We re-wrote the introduction in a more cohesive manner. - We added a small comparison between the SK and SYK models. - We added a new appendix containing the derivation of the semi-classical limit of the echo observable and the square commutator. - We added a new appendix containing the derivation of TWA and its validity for the SK model. - We re-wrote the introduction to TWA in Section 5. - We changed the caption of the pictures and added ordinate labels when missing. In Fig.4, we added an exponential function to guide the reader's eyes to the thermodynamic limit. - We improved the discussion of the numerical findings in Section 6.

Attachment:

ScraSkSemiClass_SciPost_v7_sp_compressed.pdf

Rodolfo Jalabert  on 2020-02-27  [id 747]

(in reply to Silvia Pappalardi on 2020-02-17 [id 739])
Category:
answer to question

Second report on the manuscript “Quantum echo dynamics in the Sherrington-Kirkpatrick model”
by S. Pappalardi, A. Polkovnikov, and A. Silva

I appreciate the improvements made in the presentation and discussion of the manuscript, and I think that it is now publishable in SciPost Physics.

I have nevertheless a few comments and optional suggestions.

In answering the point 1) of my original report, the authors claim that the echo observable $\mu(t)$ can be refereed to as OTOC, like “all the time-dependent correlation functions which have an unusual time-ordering”. I was not questioning the “legal right” that the authors claim to have in choosing such nomenclature. But it’s just not common sense to induce such a confusion by using a name that means something different for almost all practitioners. If the authors were to define the product of an arbitrary even or odd number of operators at various times, would they also call this object OTOC? They dubbed “square commutator” what everybody else calls OTOC. But such a choice is also questionable within their line of thought, since “square commutator” does not carry the notion of time, and could eventually refer to the square of any pair of commutators (not necessarily at different times).

The authors ask if the “Referee could kindly point” … “the reference where …". Such a discussion can be found in Ref. [18] of the latest version of the manuscript, as well as in some of its references, i.e. Phys. Rev. Lett. 121, 210601 (2018). However, please notice that nor in my original report, nor thought this answer, I am requesting to increase the citation list with rather well-known concepts.

Concerning my points 2) and 3), the authors state in the manuscript and in the response that “whenever the classical limit is chaotic” … the square commutator, and then $\mu(t)$, are expected to grow exponentially in time because they encode “the square of the derivatives of the classical trajectory to respect to the initial conditions”. However, a quantum system might have a classical analog, while the derivatives of the classical trajectory with respect to the initial condition might not be exponential because the dynamics is not fully chaotic. Therefore, having a classical analog is not a sufficient condition for observing an intermediate time-window of exponential growth. Moreover, it is not obvious to me the identification made between a system having a semi-classical analogue and the existence of a classical limit for the “square commutator” (the latter based on the validity of the saddle point mean-field approximation ensured by the large N-limit). Some spin ½ chains are shown to fulfill the second definition, but none of them fulfils the first one.

The tutorials about the Bopp formalism, included in Sec. 5 of the manuscript, in Appendix A, and in the response to Referee I, could be indeed helpful. But, other than the correction later implemented by one of the authors, I notice that in the paragraph before Eq. (23), $\alpha$ and $\beta$ are not boson operators, but complex phase-space variables, that in Ref. [70] “Vacational” has to be changed to “Variational”, and that Ref. [71] deals with the non-linear $\sigma$-model.

Author:  Silvia Pappalardi  on 2020-02-17  [id 738]

(in reply to Report 2 by Rodolfo Jalabert on 2019-12-05)
Category:
answer to question
validation or rederivation

We are grateful to the Referee for her/his careful reading of the manuscript and for pointing out the potentially broad impact of our results.

A new version manuscript has been already resubmitted on the Arxiv, where we have made few changes according to the Referees comments. However, we attach here a copy where the most relevant changes are marked in blue.

In the following, we provide detailed answers to the Referee's specific comments.

Question (1):

We thank the referee for this comment. Since a similar issue has been raised also by the Referee 1, we have added few lines in the introduction to the new version of the manuscript, improved the discussion in Section 2 and included a new small appendix with the details of the classical limit of the echo observables.

First of all, let us specify the general meaning that one can associate to OTOC. Even if in recent literature the name OTOC is usually related to correlators as $\langle \hat A \hat B(t) \hat A \hat B(t) \rangle$, ``out of time-ordered correlators'' OTOC can be referred to all the time-dependent correlation functions which have an unusual time-ordering. Namely, they describe correlation functions which cannot be computed with conventional techniques that assume casual time evolution and therefore need to be evaluated via extended Keldysh contours, like the one developed in [Aleiner, Faoro, Ioffe, Annals of Physics 2016]. In this sense, the echo observable $\mu(t)$ can be refereed as OTOC, since it contains terms as $\langle \hat B(t) \hat A \hat B(t) \rangle$, see Eq.(4).

In the second place, in the semi-classical limit, every kind of OTOC contains quadratic terms in the derivatives of the initial conditions, as was already known by Larkin and Ovchinnikov. This is true also for the echo $\mu(t)$. This statement can be shown by the use of time-dependent Bopp operators, a standard formalism of phase-space methods which allows computing time-dependent correlation functions, by representing quantum operators as phase-space variables. This is extensively discussed in the Appendix of [Schmitt, Sels, Kehrein and Polkovnikov, PRB 2019] for a semi-classical limit of the SYK. Anyhow, in order to be as self-consistent as possible, we have added a new Appendix to the manuscript, with the derivation of the semi-classical limit of $\mu(t)$, based on Bopp operators. We show for bosonic and spin operators, that the semi-classical echo observable contains quadratic terms in the derivatives of the classical trajectory to respect to the initial conditions, hence it encodes the Lyapunov exponent, exactly as the square commutator. For further details on this derivation, we refer to the response (1) to the Report 1, where we have reported the main steps of the proof.

In conclusion, we acknowledge that the square commutator $c(t) = - \langle [\hat B(t), \hat A]^2 \rangle$ contains a term $\langle B(t)A^2 B(t)\rangle$, which is missed by $\mu(t)$. For this reason, we added a sentence in Section 2. Since we find this very interesting, we would like to ask if the Referee could kindly point us the reference where ``the terms that make up for their difference have been shown to be crucial in obtaining the exponential growth of the former as the difference between parametrically larger components.''

Question (2):

We thank the referee for this comment since this point is indeed a non-trivial statement and it is exactly one of the purposes of the paper. If we believe that there are no other initial time scales, our claim can be re-stated as follows: If there exist a time-interval $t^\ast\leq t \leq t_{\text{Ehr}}$, where $t_{\text{Ehr}}$ diverges in the thermodynamic limit, then in the same interval the $\mu(t)$ should be semi-classical. Therefore it behaves like the square of the derivatives to respect to the initial conditions, see also the Refs.[16-18] of the manuscript and the new Appendix A. When the latter grows exponentially with a rate given by the classical (generalized) Lyapunov exponent, then - in that time interval - also the quantum $\mu(t)$ should grow exponentially with the same rate. On the other hand, when the system does not have a semi-classical analogue, i.e. $t_{\text{Ehr}}\sim t^*$ does not depend on the system size, there is no time-regime where the quantum $\mu(t)$ would behave semi-classically. Therefore there is no reason in principle to have exponential growth. This is typically the case of short-range interacting lattice systems, which do not have a classical analogue and for which it has been proved that the square-commutator grows at most exponentially fast [Kukulian et al.]. This statement is firmly supported by our numerics on long-range (semi-classical) and short-range (fully-quantum) models.

** Question (3):**

We thank the Referee for this comment since our previous version did not provide the underlying argument to our statements. We have added a discussion and a new appendix in order to better justify our claims. The justification can be summarized as follows. TWA can be proved as the saddle point of the Keldysh action in a path-integral formulation of the time-evolutions. Within this framework, representation of initial conditions through the Wigner function is exact, while the approximation occurs at the level of the time-evolution, which is solved by the saddle-point approximation expanding in small quantum fluctuations. Even if the present work deals with spins $1/2$, this all-to-all interacting model has a mean-field character and the large $N$-limit ensures the validity of the saddle point approximation, as confirmed by the numerics. More formally (see Appendix B), one can estimate the corrections to the saddle-point solution, which are found to scale as $1/N$.

Question (4):

We thank the referee for highlighting this point, since the presence of non-local interactions, shared by the SYK and SK models, is indeed crucial to have a semi-classical limit and hence to observe the exponential growth of the OTOC. Even if in the previous version we already pointed out that the models share several analogies, we have now added a sentence to signal that the SYK and SK differ in many respects. For example, they have different equilibrium phase-diagrams, which inevitably lead to differences in the dynamics. This is particularly evident at low temperatures, where the SK model possesses a glassy phase below a critical transverse field, i.e. see [Andreanov and Muller, PRL 2012]. On the other hand, such a phase-transition is absent in the SYK model. Anyhow, as pointed out by the Referee, the two models share the feature of having non-local all-to-all random interaction. Indeed, a semi-classical analysis was performed by one of us on the SYK using the TWA on fermionic bilinears, see [Schmitt, Sels, Kehrein and Polkovnikov, PRB 2019], where the authors found similar results concerning the exponential growth of the echo. In that paper, the role of the short-range interactions was not addressed, and the analysis presented in the present work shows how the non-locality of the interactions affects the classical limit (even on a system of spins $1/2$) and therefore the exponential growth. Also in that paper there was no analysis of the observables and the echo operator and the choice, which satisfies the criteria we derived here was rather coincidental. We believe that finding these conditions is another important new result of the present work.

Question (5):

We changed the figure's captions and added the ordinate axis where missing. We made an effort to improve the presentation, correcting misspellings and rewriting some sections.

---

## Round 2 · Referee Report · Anonymous · 2020-2-28

Strengths

1-timely subject (OTOCs)
2-cutting edge analytics and semiclassics
3-Relevant results about time-scales and possibility to observe exponetial growth of the OTOC.

Weaknesses

1-OTOC considered is not the most widespread quantity

Report

The authors have answered the concerns of the referees rather carefully and I think the manuscript in the present form is a significant contribution to the subject of characterizing chaos in complex quantum systems. I recommend publication.

---

## Round 2 · Author Response

Dear Editor,

thank you for handling our submission.

We are pleased to thank the referees, whose insightful comments and observations helped us further improve and clarify our work. In this resubmission, we believe to have addressed all the point raised by the Referees. We append below a detailed response to the reports and the list of changes.

Yours sincerely,

Silvia Pappalardi, Anatoli Polkovnikov and Alessandro Silva

Reply to referee report I:

We thank the referee for her/his thoughtful reading on the manuscript. Here we reply to the corresponding points of their report, specifying the changes included in the resubmitted manuscript.

1) We thank the referee for the question since our previous draft lacked clarity. Since a similar issue has been raised also by the Referee 2, we have added few lines in the introduction, improved the discussion in Section 2 and included a new small appendix with the details of the classical limit of the echo observables.

We now explain in details the relation between $\mu(t)$ and OTOCs. First of all, let us specify the general meaning that one can associate to OTOC. Even if in recent literature the name OTOC is usually related to correlators as $\langle \hat A \hat B(t) \hat A \hat B(t) \rangle$, "out of time-ordered correlators" OTOC can be referred to all the time-dependent correlation functions which have an unusual time-ordering. Namely, they describe correlation functions which cannot be computed with conventional techniques that assume casual time evolution and therefore need to be evaluated via extended Keldysh contours, like the one developed in [Aleiner, Faoro, Ioffe, Annals of Physics 2016]. In this sense, the echo observable $\mu(t)$ can be refereed as OTOC, since it contains terms as $\langle \hat B(t) \hat A \hat B(t) \rangle$, see Eq.(4).

In the second place, in the semi-classical limit, every kind of OTOC contains quadratic terms in the derivatives of the initial conditions, as was already known by Larkin and Ovchinnikov. This is true also for the echo $\mu(t)$. This statement can be shown by the use of time-dependent {Bopp operators}, a standard formalism of phase-space methods which allows computing time-dependent correlation functions, by representing quantum operators as phase-space variables. This is extensively discussed in the Appendix of [Schmitt, Sels, Kehrein and Polkovnikov, PRB 2019] for a semi-classical limit of the SYK. In order to be as self-consistent as possible, we have added a new Appendix to the manuscript, with the derivation of the semi-classical limit of $\mu(t)$, based on Bopp operators. We show for bosonic and spin operators, that the semi-classical echo observable contains quadratic terms in the derivatives of the classical trajectory to respect to the initial conditions, hence it encodes the Lyapunov exponent, exactly as the square commutator. In what follows, we report the core of the proof.

Let us start by introducing the Bopp formalism. The Wigner-Weyl quantization is intrinsically connected with the symmetric Bopp representation of the quantum operators, which allows to map operators to functions of phase space variables. In particular, the bosonic creation and annihilation operators in the Bopp representation read

\[ \hat a^\dagger \to \alpha^\ast-{1\over 2} {\partial\over \partial\alpha},\quad \hat a\to\alpha+{1\over 2} {\partial\over \partial \alpha^\ast}. \quad \quad (1) \]

Then the Weyl symbol of, for example, the number operator is obtained by simply writing it in the Bopp representation $n^w=(\hat a^\dagger \hat a)^w=\left (\alpha^\ast-{1\over 2} {\partial\over \partial\alpha}\right)\alpha=\alpha^\ast\alpha-{1\over 2}$. Semi-classical expectation values are obtained by averaging the Weyl symbol of the operator with the Wigner function corresponding the the quantum state (see also the new Section 5 of the manuscript). Interestingly the Bopp representation immediately allows one to compute non-equal correlation functions e.g.

\[ \left (\hat a^\dagger(t_1) \hat a(t_2) \right )_w = \alpha^\ast (t_1) \alpha (t_2)-{1\over 2}{\partial \alpha(t_2)\over\partial \alpha(t_1)} \ ,\quad \quad (2) \]

where the derivative to respect to $\alpha(t_1)$ represents the non-equal time response. One can also write the Bopp operators in a more compact form where the creation and annihilation operators (and similarly the momentum and the coordinate operators) map to the corresponding phase space variables plus half of the Poisson bracket. For more complicated operators, like non-linear bosonic variables or spin operators, this simple interpretation is lost as generally higher order derivatives emerge. In order to derive the semi-classical limit of OTOC at order $\hbar^2$, it is enough to keep at most the second-order expansion in $\hbar$ of the Bopp operator. In particular, for a generic time-dependent operator $\hat B(t)$, the Bopp representation can be written as

\[ \hat B(t) \to B_t + \hbar D^{(1)}_{B_t} + \hbar^2 D^{(2)}_{B_t} \ ,\quad \quad (3) \]

where $B_t$ is the Weyl symbol of the operator $\hat B$ evaluated at time $t$, the linear order is given by half of the Poisson brackets
$D_{B_t}^{(1)}=i/2${$B_t, \cdot$}, and $D^{(2)}_{B_t}$ contains the second-order derivatives and its explicit form depends on the operator $\hat B(t)$. These formulae can be used in constructing the Weyl symbols for various time-dependent expectation values see [Polkovnikov Ann.of.Phys. (2010)] and, in particular, to compute out-of-time ordered correlators.

To do so, we consider the Bopp representation of $\hat B(t)$ [cf. Eq.(3)] and the corresponding one for $ \hat A(0) \to A_0 + \hbar D_{A_0}^{(1)}+ \hbar^2 D_{A_0}^{(2)} $.

To compute the semi-classical limit the echo observable and the square commutator, we evaluate the Weyl symbol of several correlation functions, e.g.

\[ \left( \hat B(t) \, \hat A \, \hat B(t) \, \right)_w = (B_t + D^{(1)}_{B_t} + D_{B_t}^{(2)})\, (A_0 + D^{(1)}_{A_0} + D_{A_0}^{(2)})\, B_t \ , \]

and we simplify the resulting expressions. After a tedious calculation, the Weyl symbol of the echo observable $\mu(t) = -1/2 \langle [\hat B(t), [\hat B(t), \hat A]]\rangle $ reads

\[ \left ( [ \hat B(t), [\hat B(t), \hat A(0)]\, ]\right )_w = \hbar^2\left [3 \left(D_{B_t}^{(1)} \right )^2 A_0 - D_{B_t}^{(2)}\, B_t A_0 + D_{A_0}^{(2)} B_t^2 + A_0 \, D_{B_t}^{(2)} B_t \right ]\ , \quad \quad (4) \]

while for the square commutator $c(t) = - \langle [ \hat B(t), \hat A(0)] \, ]^2\rangle$ one finds

\[ -\left ( [ \hat B(t), \hat A(0)]\, ]^2 \right )_w = - 4 \hbar^2\, \left ( D_{A_0}^{(1)} B_t \right )^2 = \hbar^2\, \{ A_0, B_t\}^2 \ . \quad \quad (5) \]

It is well known that the classical limit of the square commutator [cf. Eq.(5)] encodes the square of the derivatives of the classical trajectory to respect to the initial conditions (see Refs.[48-50] in the manuscript). This means that, whenever the classical limit is chaotic, $c(t)$ is expected to grow exponentially, with a rate given by twice the largest Lyapunov exponent. Let us consider a simple example by choosing $\hat A(0) = \hat a^2(0)$ and $\hat B(t) = \hat a^{\dagger}(t)$, for which

\[ \begin{align*} B_t& =\alpha_t^\ast\ , \quad D^{(1)}_{B_t} = - \frac 1{2} \, \frac{\partial }{\hbar \partial \alpha_t} \ , \quad D^{(2)}_{B_t}=0 \\ A_0& =\alpha^2(0)\ , \quad D^{(1)}_{A_0} = {\alpha(0)}\frac{\partial}{ \hbar\partial \alpha(0)^{\ast}}\ , \quad D^{(2)}_{A_0}=\frac 1{4} \frac{\partial^2}{\hbar^2 \partial \alpha(0)^{\ast\, 2}} \ . \end{align*} \]

It is then easy to show that Eq.(5) simply gives $ c(t) \to -4 \alpha^2(0)\left( \frac{ \partial \alpha^\ast(t)}{ \partial \alpha^{\ast}(0)} \right)^2 $. We now show that the same result applies to the semi-classical limit of the echo observable (4), for instance considering the previous example. Substituting the Bopp representation for $\hat A(0) = \hat a^2(0)$ and $\hat B(t) = \hat a^{\dagger}(t)$ into Eq.(4), and using the chain rule for the second-order derivatives, one gets

\[ \left ( [ \hat a^{\dagger}(t), [\hat a^{\dagger}(t), \hat a^2(0)]\, ]\right )_w = \frac \hbar2 \left [ 3 \left ( \frac{\partial\alpha(0)}{\partial \alpha(t)}\right )^2 + \left ( \frac{\partial\alpha^*(t)}{\partial \alpha^*(0)}\right )^2 + 3 \alpha(0)\frac{\partial^2 \alpha(0)}{\partial \alpha^2(t)} + \alpha^\ast(t)\frac {\partial^2 \alpha^{\ast}(t)}{\partial \alpha^{\ast\, 2}(0)} \right ] \ , \]

which, exactly as the square commutator, is dominated by the square of the derivatives of the classical trajectory to respect to the initial conditions. In the new Appendix B, we discuss also the Bopp representation of spin variables $\hat A=\hat B = \hat S^z$ and we show that, also in this case, the semiclassical echo observable is proportional to the square of the derivatives of classical spin trajectory $S^z_t$ to respect to the initial conditions $S_0^{x,y,z}$.

2) We thank the referee for the question and we acknowledge that the previous version of the manuscript was intended to readers who already had familiarity with the method. First of all, to improve the explanation of the TWA, we have added in the new version a broader introduction of the method, considering as phase-space example the coherent state representation. We explained how to use it to compute time-dependent expectation values and multi-time correlation functions. Furthermore, since TWA can be derived as the saddle point approximation of the Keldysh path-integral, we have added a new Appendix in which we sketch the steps of the proof. In the second place, we better discuss the justification of TWA for the SK model comes from the validity of this saddle point approximation due to the mean-field nature of the fully-interacting SK model. We made an effort also to justify this both in the appendix and in the section devoted to the TWA on the SK.

3) We thank the referee for this question since a large part of our results is based on the use of TWA. We have added the following discussion in the new version of the manuscript. As we have argued in the paper, the quantum exponential growth is restricted in the time interval ${t^* \leq t \leq t_{\text{Ehr}}\sim\log N}$. Therefore, in order to fully appreciate it numerically, one would need to simulate very big system sizes, being $\log N$ a very slow function of its argument. Since exact diagonalization is limited to very small system sizes, e.g. $[\log(8):\log(20)] \sim [0.9: 1.3]$, a different numerical approach is needed to reproduce the behaviour for large $N$. As we show numerically in Section 5.1 and now justify in Appendix B, for the model under analysis TWA meets this need, reproducing the exact quantum dynamics for big $N$ before the Ehrenfest time $t_{\text{Ehr}}$. Furthermore, TWA is more accurate in extracting the Lyapunov exponent $\Lambda$ for the additional two reasons: 1) TWA does not know about the Ehrenfest time (a fully quantum time-scale) and its exponential growth lasts for many decades. 2) In TWA $\Lambda$ becomes independent on the system size even for relatively small $N$, allowing a precise estimate. Fig.5 is used to show the usefulness of this approach. The ED data grow exponentially only at short times where they coincide with TWA (before $t\sim1.5$), see also Fig.4 (i). While the TWA data continue to grow - with the same rate - for a few decades allowing to accurately extract $\Lambda$. To summarize this discussion the TWA for the echo indeed breaks down at relatively short times unless $N$ is huge. But it breaks down in a smart way predicting what quantum dynamics would look like of $N$ becomes exponentially large. While this result seems to be paradoxical it is correct and not incidental. By our arguments, it should apply to any large N model, which has a diverging Ehrenfest time. This loosely follows from the fact that the main role of $N$ in dynamics is to set the value of $\hbar$, other corrections due to finite $N$ are small an very quickly disappear as $N$ becomes moderately large, of the order of 10. So the semiclassical-classical TWA dynamics effectively extrapolate $\hbar\to 0$ and is very efficient if we are interested in this limit.

4) Reply to the minor comments:

  • We have added the citations.

  • We have added further references.

  • We thank the referee for pointing out a mistake in the notations. Generally, if one assumes that the time-evolution of a quantum state with an hamiltonian $\hat H$ is $e^{-i \hat H t}\ket{\psi_0}$, then the Heisenberg representation for the operator $\hat A$ is $\hat A(t) = e^{i \hat H t} A e^{-i \hat Ht}$. Anyhow, right after Eq.(1), it was stated that \emph{``$\hat B(t) = e^{ -i\hat H t} \hat B \, e^{ i\hat H t}$ is the perturbing operator in the Heisenberg representation with respect to the forward Hamiltonian $\hat H$''}, which is therefore not correct. We correct this by first performing the evolution with the backward hamiltonian $-H$, then we consider the small rotation and then evolve with the reversed hamiltonian $\hat H.$

  • We explained in what sense Eq.(2) contains the OTOC, see reply to the point (1).

  • We added references.

  • We have added justification for the use of TWA for the SK model in a new Appendix B and made an effort to make the section more clear.

  • We removed the `and hence the otoc' since it is not necessary for this part.

Reply to referee report II:

We thank the referee for her/his interesting comments. Here we reply to the corresponding points of their report, specifying the changes included in the resubmitted manuscript.

1) We thank the referee for its comment. Since a similar issue has been raised also by the Referee 1, we have added few lines in the introduction, improved the discussion in Section 2 and included a new small appendix with the details of the classical limit of the echo observables.

First of all, let us specify the general meaning that one can associate to OTOC. Even if in recent literature the name OTOC is usually related to correlators as $\langle \hat A \hat B(t) \hat A \hat B(t) \rangle$, "out of time-ordered correlators" OTOC can be referred to all the time-dependent correlation functions which have an unusual time-ordering. Namely, they describe correlation functions which cannot be computed with conventional techniques that assume casual time evolution and therefore need to be evaluated via extended Keldysh contours, like the one developed in [Aleiner, Faoro, Ioffe, Annals of Physics 2016]. In this sense, the echo observable $\mu(t)$ can be refereed as OTOC, since it contains terms as $\langle \hat B(t) \hat A \hat B(t) \rangle$, see Eq.(4). In the second place, in the semi-classical limit, every kind of OTOC contains quadratic terms in the derivatives of the initial conditions, as was already known by Larkin and Ovchinnikov. This is true also for the echo $\mu(t)$. This statement can be shown by the use of time-dependent {Bopp operators}, a standard formalism of phase-space methods which allows computing time-dependent correlation functions, by representing quantum operators as phase-space variables. This is extensively discussed in the Appendix of [Schmitt, Sels, Kehrein and Polkovnikov, PRB 2019] for a semi-classical limit of the SYK. Anyhow, in order to be as self-consistent as possible, we have added a new Appendix to the manuscript, with the derivation of the semi-classical limit of $\mu(t)$, based on Bopp operators. We show for bosonic and spin operators, that the semi-classical echo observable contains quadratic terms in the derivatives of the classical trajectory to respect to the initial conditions, hence it encodes the Lyapunov exponent, exactly as the square commutator. For further details on this derivation, we refer to the response (1) to the Report 1, where we have reported the main steps of the proof.

In conclusion, we acknowledge that the square commutator $c(t) = - \langle [\hat B(t), \hat A]^2 \rangle$ contains a term $\langle B(t)A^2 B(t)\rangle$, which is missed by $\mu(t)$. For this reason, we added a sentence in Section 2. Since we find this very interesting, we would like to ask if the Referee could kindly point us the reference where "the terms that make up for their difference have been shown to be crucial in obtaining the exponential growth of the former as the difference between parametrically larger components".

2) We thank the referee for this comment since this point is indeed a non-trivial statement and it is exactly one of the purposes of the paper. If we believe that there are no other initial time scales, our claim can be re-stated as follows: If there exist a time-interval $t^{\ast}\leq t \leq t_{\text{Ehr}}$, where $t_{\text{Ehr}}$ diverges in the thermodynamic limit, then in the same interval the $\mu(t)$ should be semi-classical. Therefore it behaves like the square of the derivatives to respect to the initial conditions, see also the Refs.[16-18] of the manuscript and the new Appendix A. When the latter grows exponentially with a rate given by the classical (generalized) Lyapunov exponent, then - in that time interval - also the quantum $\mu(t)$ should grow exponentially with the same rate. On the other hand, when the system does not have a semi-classical analogue, i.e. $t_{\text{Ehr}}\sim t^{\ast}$ does not depend on the system size, there is no time-regime where the quantum $\mu(t)$ would behave semi-classically. Therefore there is no reason in principle to have exponential growth. This is typically the case of short-range interacting lattice systems, which do not have a classical analogue and for which it has been proved that the square-commutator grows at most exponentially fast [Kukulian et al.]. This statement is firmly supported by our numerics on long-range (semi-classical) and short-range (fully-quantum) models.

3) We thank the Referee for this comment since our previous version did not provide the underlying argument to our statements. We have added a discussion and a new appendix in order to better justify our claims. The justification can be summarized as follows. TWA can be proved as the saddle point of the Keldysh action in a path-integral formulation of the time-evolutions. Within this framework, representation of initial conditions through the Wigner function is exact, while the approximation occurs at the level of the time-evolution, which is solved by the saddle-point approximation expanding in small quantum fluctuations. Even if the present work deals with spins $1/2$, this all-to-all interacting model has a mean-field character and the large $N$-limit ensures the validity of the saddle point approximation, as confirmed by the numerics. More formally (see Appendix B), one can estimate the corrections to the saddle-point solution, which are found to scale as $1/N$.

4) We thank the referee for highlighting this point, since the presence of non-local interactions, shared by the SYK and SK models, is indeed crucial to have a semi-classical limit and hence to observe the exponential growth of the OTOC. Even if in the previous version we already pointed out that the models share several analogies, we have now added a sentence to signal that the SYK and SK differ in many respects. For example, they have different equilibrium phase-diagrams, which inevitably lead to differences in the dynamics. This is particularly evident at low temperatures, where the SK model possesses a glassy phase below a critical transverse field, i.e. see [Andreanov and Muller, PRL 2012]. On the other hand, such a phase-transition is absent in the SYK model. Anyhow, as pointed out by the Referee, the two models share the feature of having non-local all-to-all random interaction. Indeed, a semi-classical analysis was performed by one of us on the SYK using the TWA on fermionic bilinears, see [Schmitt, Sels, Kehrein and Polkovnikov, PRB 2019], where the authors found similar results concerning the exponential growth of the echo. In that paper, the role of the short-range interactions was not addressed, and the analysis presented in the present work shows how the non-locality of the interactions affects the classical limit (even on a system of spins $1/2$) and therefore the exponential growth. Also in that paper, there was no analysis of the observables and the echo operator and the choice, which satisfies the criteria we derived here was rather coincidental. We believe that finding these conditions is another important new result of the present work.

5) We changed the figure's captions and added the ordinate axis where missing. We made an effort to improve the presentation, correcting misspellings and changing some words.

---

## Round 2 · List of Changes

- We re-wrote the introduction in a more cohesive manner.
- We added a small comparison between the SK and SYK models.
- We added a new appendix containing the derivation of the semi-classical limit of the echo observable and the square commutator.
- We added a new appendix containing the derivation of TWA and its validity for the SK model.
- We re-wrote the introduction to TWA in Section 5.
- We changed the caption of the pictures and added ordinate labels when missing. In Fig.4, we added an exponential function to guide the reader's eyes to the thermodynamic limit.
- We improved the discussion of the numerical findings in Section 6.
- Updated bibliography.

---

## Round 3 · Author Response

Errors in user-supplied markup (flagged; corrections coming soon)

Dear Editor,

thank you for handling our submission. We apologize for the delay in this resubmission.
We are pleased to thank the referees for recommending the publication of the paper on SciPost Physics.
In this resubmission, we believe to have addressed the final comments and suggestions raised by Referee 2.

Yours sincerely,

Silvia Pappalardi, Anatoli Polkovnikov, and Alessandro Silva
* * *
Reply to the Report of Prof. Jalabert:
* * *
We thank the referee for the comments and careful reading. Here, we reply to the corresponding points, specifying the changes included in the resubmitted manuscript.

**The referee writes:**
> In answering the point 1 of my original report, the authors claim that the echo observable $\mu(t)$ can be referred to as OTOC, like “all the time-dependent correlation functions which have an unusual time-ordering”. I was not questioning the “legal right” that the authors claim to have in choosing such nomenclature. But it’s just not common sense to induce such confusion by using a name that means something different for almost all practitioners. If the authors were to define the product of an arbitrary even or an odd number of operators at various times, would they also call this object OTOC? They dubbed “square commutator” what everybody else calls OTOC. But such a choice is also questionable within their line of thought, since “square commutator” does not carry the notion of time, and could eventually refer to the square of any pair of commutators (not necessarily at different times).

**Our response:**

Following the suggestion of the referee, we have changed the sentences where we refer to the echo $\mu(t)$ as *the* OTOC, but we kept the one in the introduction where we say contains *an* OTOC. In fact, while we agree about the fact that all practitioners define $\langle \hat B(t) \hat A(0) \hat B(t) \hat A(0)\rangle$ as *the* OTOC, we still believe that objects as $\langle \hat B(t) \hat A(0) \hat B(t) \rangle$ lie in the same class, sharing the unusual time-ordering and a similar classical limit.
More precisely, OTOC are defined as multi-point and multi-time correlation functions (more or equal than three body) which cannot be represented on a single Keldysh contour, following the work by Alainer, Faoro, Ioffe, Annals of Physics (2016). They are characterized by an unusual time-ordering which prevents them from appearing in standard causal response functions. In order to be more accurate, we have added this definition in the introduction of the revised manuscript.
Furthermore, for the particular initial state we are considering, i.e. $\hat A|\psi_0 \rangle = \alpha_0 |{\psi_0}\rangle$, the OTOC appearing in the echo $\mu(t)$ is equal to the ``standard'' OTOCs, by a constant factor, i.e.

$$
\langle \hat B(t) \hat A(0) \hat B(t) \rangle
= \frac 1{\alpha_0} \langle \hat B(t) \hat A(0) \hat B(t) \hat A(0)\rangle
$$

To conclude, we would like to reiterate that our motivation for choosing the echo $\mu(t)$ is very similar to what discussed by Boris Fine and collaborators in Refs.[23-24]. There, they propose the echo as a simple and easy way to measure observables and encode irreversible dynamics via some particular OTOC.

**The referee writes:**
>Concerning my points 2 and 3, the authors state in the manuscript and in the response that “whenever the classical limit is chaotic” … the square commutator, and then $\mu(t)$, are expected to grow exponentially in time because they encode “the square of the derivatives of the classical trajectory to respect to the initial conditions”. However, a quantum system might have a classical analogue, while the derivatives of the classical trajectory with respect to the initial condition might not be exponential because the dynamics are not fully chaotic. Therefore, having a classical analogue is not a sufficient condition for observing an intermediate time-window of exponential growth. Moreover, it is not obvious to me the identification made between a system having a semi-classical analogue and the existence of a classical limit for the “square commutator” (the latter based on the validity of the saddle point mean-field approximation ensured by the large N-limit). Some spin $1/2$ chains are shown to fulfil the second definition, but none of them fulfils the first one.

**Our response:**
We thank the referee for raising this point. More precisely, we meant that every time the classical limit is well defined, all quantum observables (including the echo $\mu(t)$) shall display semi-classical dynamics before $t_{\text{Ehr}}$. At this point, if the classical limit is sufficiently chaotic, i.e. the derivatives of the trajectory to respect the initial condition immediately grow exponentially fast, then also the quantum echo observable should reproduce such exponential growth.
In the example of the SK model under consideration, the classical limit is fully chaotic as witnessed by the exponential growth of the echo within TWA. Hence, one could predict that the quantum $\mu(t)$ should grow exponentially fast in time before $t_{\text{Ehr}}$. Following the comment of the referee, in the revised manuscript, we have specified *chaotic* when referring to the semi-classical limit as a condition for the exponential growth.

Let us now briefly comment on the second part of the referee's observation.
As the referee points out, typically quantum spin 1/2 chains with short-range interactions do not have a classical limit, either semiclassical correspondence of the square-commutator. However, in the case we are studying, the validity of the semiclassical limit is ensured by the presence of long-range interactions and in particular of the large $N$ limit. Even if this is not an "obvious'' classical limit (like it would be for large $S$ spin chains), anyways the large $N$ limit is enough for having a semi-classical description and semi-classical dynamics.
Physically, this follows from the fact that each spin is subject to a slowly changing magnetic field generated by interactions with many other spins. So effectively each spin evolves in an external field, for which the semiclassical description is exact. This semiclassical behaviour is manifested both in the dynamics of local observables (Figure 1 and 2) and in more complicated observables, like the echo (Figure 4). This has been shown numerically in Section 5 and analytically in appendix B.

**The referee writes:**
>The tutorials about the Bopp formalism, included in Sec. 5 of the manuscript, in Appendix A, and in the response to Referee I, could be indeed helpful. But, other than the correction later implemented by one of the authors, I notice that in the paragraph before Eq. (23), $\alpha$ and $\beta$ are not boson operators, but complex phase-space variables, that in Ref. [70] “Vacational” has to be changed to “Variational”, and that Ref. [71] deals with the non-linear -model.

**Our response:**
We would like to thank the referee for pointing out these mistakes. We have corrected them in the resubmitted version of the manuscript.

---

## Round 3 · List of Changes

- We added the definition of OTOC in the introduction as a "multi-point and multi-time correlation functions which cannot be represented on a single Keldysh contour".
- We have specified *chaotic* when referring to the semi-classical limit as a condition for the exponential growth.
- We have corrected the typos in the references [70-71].
- We implemented the corrections to the Bopp formalism.

---

## Editorial Decision

published